# Multimodal deep learning integration of cryo-EM and AlphaFold3 for high-accuracy protein structure determination
Rajan Gyawali [1,2], Ashwin Dhakal[1,2] & Jianlin Cheng [1,2] ✉

Cryo-electron microscopy (cryo-EM) is a key technology for determining the structures of proteins, particularly large protein complexes. However, automatically building high-accuracy protein structures from cryo-EM density maps remains a crucial challenge. In this work, we introduce MICA, a fully automatic and multimodal deep learning approach combining cryo-EM density maps with AlphaFold3-predicted structures at both input and output levels to improve cryo-EM protein structure modeling. It first uses a multi-task encoder-decoder architecture with a feature pyramid network to predict backbone atoms, Cα atoms, and amino acid types from both cryo-EM maps and AlphaFold3-predicted structures, which are used to build an initial backbone model. This model is further refined using AlphaFold3-predicted structures and density maps to build final atomic structures. MICA significantly outperforms other state-of-the-art deep learning methods in terms of both modeling accuracy and completeness, and is robust to protein size and map resolution. Additionally, it builds high-accuracy structural models with an average template-based modeling score (TM-score) of 0.93 from recently released high-resolution cryo-EM density maps, showing it can be used for real-world, automated, accurate protein structure determination.

Understanding the three-dimensional (3D) atomic structure of proteins is fundamental to structural biology, as it provides key insights into protein function and has wide applications in many biological and medical domains, such as drug and vaccine development[1–3]. Cryo-EM has evolved as a leading technology for resolving protein structures at atomic or near-atomic resolution[4–6]. It is currently the most prominent technique for determining the structures of large protein complexes and assemblies[7–12]. The advancement of cryo-EM imaging and protein particle picking[13–16] has enabled cryo-EM to generate more and more high-resolution density maps (<4 Å) for proteins that can be used to build atomic protein structures. However, building high-accuracy protein structures from high-resolution cryo-EM density maps remains challenging and usually requires extensive intervention by human experts.

The primary hurdles in this modeling process involve accurately identifying atoms within the density maps, tracing these atoms to construct protein backbone structures, and aligning amino acid sequences with them[9]. Traditional molecular modeling tools like Rosetta[17], MAINMAST[18], and Phenix[10] made significant progress, but they still require extensive manual intervention, rely heavily on structural templates, and produce incomplete, fragmented protein structures. These limitations underscore the need for fully automated solutions to reconstruct high-accuracy protein structures from cryo-EM density maps.

In response to these challenges, new machine learning methods have been introduced to automate cryo-EM protein structure modeling over the last several years[19]. DeepTracer[9], for instance, offers an automated approach that uses a convolutional neural network (CNN) to determine atomic protein structures from cryo-EM maps, considerably improving speed and accuracy over previous automated methods. Cryo2Struct[12] provides a fully automated, ab initio solution that generates 3D atomic structures directly from cryo-EM density maps without relying on any predicted or homologous template structures. It employs a transformer-based deep learning model to identify atoms and amino acid types in cryo-EM density maps, combined with a hidden Markov model (HMM) to align protein sequences with the atoms to build atomic structures. It improved modeling accuracy over existing de novo modeling methods.

Additionally, ModelAngelo[20] combines structural information extracted from cryo-EM maps with protein sequences and structural information captured by protein language models (i.e., ESM[21]) to improve the accuracy of model building over de novo modeling using only density maps. Recently, the incorporation of AlphaFold2 predicted structural

[1]Department of Electrical Engineering and Computer Science, University of Missouri, Columbia, MO, USA. [2]NextGen Precision Health, University of Missouri, Columbia, MO, USA. ✉e-mail: chengji@missouri.edu

models[22] into cryo-EM structure modeling, as demonstrated in DeepMainmast[23], has enhanced the accuracy of cryo-EM structure modeling. DeepMainmast blends AlphaFold2 predicted structures with de novo models built by deep learning from density maps when they exhibit similar local conformations, creating hybrid models that leverage both experimental density information and computational structure predictions.

Similarly, EModelX(+AF)[24] uses a U-Net to map carbon-alpha (Cα) positions predicted from density maps to protein sequences using profile sampling and sequence alignment techniques, where high-confidence Cα-sequence mappings are identified and registered to create an initial structure model. It is further refined by a sequence-guided threading algorithm to fill unmodeled gaps[22] using AlphaFold2 structures.

Despite the significant progress made by the recent deep learning methods in combining cryo-EM density maps with AlphaFold predicted structures, the integration of the two is at the output level, i.e., AlphaFold-predicted structures are used to improve the structures built from density maps in the final postprocessing step. We conceive that AlphaFold structures can be used with cryo-EM density maps as input for deep learning to build more accurate protein structures in the first place. The deep learning integration of cryo-EM density maps and AlphaFold-predicted structures at the input level is completely different from the current approaches of combining them at the output level, while the input-level combination and the output-level combination can be used together to maximize the accuracy of cryo-EM protein structure modeling.

In this spirit, we introduce MICA, a deep learning method for multimodal integration of cryo-EM maps and AlphaFold3 (AF3)[25] structures through an encoder-decoder architecture with a Feature Pyramid Network (FPN)[26]. The integration of the multimodal data at the input level enables MICA to leverage both cryo-EM experimental data and AF3 structures predicted from sequences, compensating for limitations inherent in each individual modality (e.g., low-resolution or missing regions in cryo-EM density maps or incorrectly predicted regions in AF3 structures). Multi-scale feature fusion through FPN captures hierarchical structural information at different resolutions, from local atomic details to global protein fold patterns, which is crucial for accurate structural modeling across diverse

protein sizes and complexities. The unified architecture with task-specific decoder heads allows simultaneous prediction of interconnected structural (e.g., backbone atoms) and sequence (amino acid type) properties within a single network. The predicted backbone atoms, Cα atoms, and amino acid types are subsequently used to build backbone structures by mapping Cα positions to protein sequences. The unmodeled or mismodeled regions in the constructed backbone structures are further refined by combining them with AF3-predicted structures in the postprocessing step to generate final structures as in EModelX( + AF)[24].

Tested on two sets of density maps with resolution between 1.5 Å and 4 Å, MICA significantly outperformed two existing state-of-the-art methods, ModelAngelo and EModelX(+AF). It also constructed high-accuracy structural models for recently released cryo-EM density maps with resolution between 2.08 Å and 3.82 Å. The results demonstrate that integrating cryo-EM density maps and AF3-predicted structures by multimodal deep learning to build atomic protein structures is a promising approach to achieving automated, highly accurate cryo-EM protein structure determination.

## Results

### An overview of multimodal deep learning integration of cryo-EM density maps and AF3 structures for atomic structure modeling

Figure 1 presents an overview of the pipeline of MICA for building protein structures. The input for MICA is the multimodal data of a protein consisting of a cryo-EM density map and AF3-predicted structures[25] of its chains along with their amino acid sequences (Fig. 1A), whereas the output is a 3D atomic structural model of the protein (Fig. 1I). MICA predicts the positions of backbone atoms, Cα atoms and amino acid types of Cα atoms from the input, which are then used with the sequences and AF3-predicted structures to build 3D atomic structures of proteins.

As shown in Fig. 1D, the features of 3D grids, extracted from cryo-EM density maps and AF3-predicted structures, are fused as input for a deep learning network (Fig. 1E) to predict backbone atoms, Cα atoms, and amino acid types. It starts with a progressive encoder stack comprising three encoder blocks with increasing feature depth to generate hierarchical feature

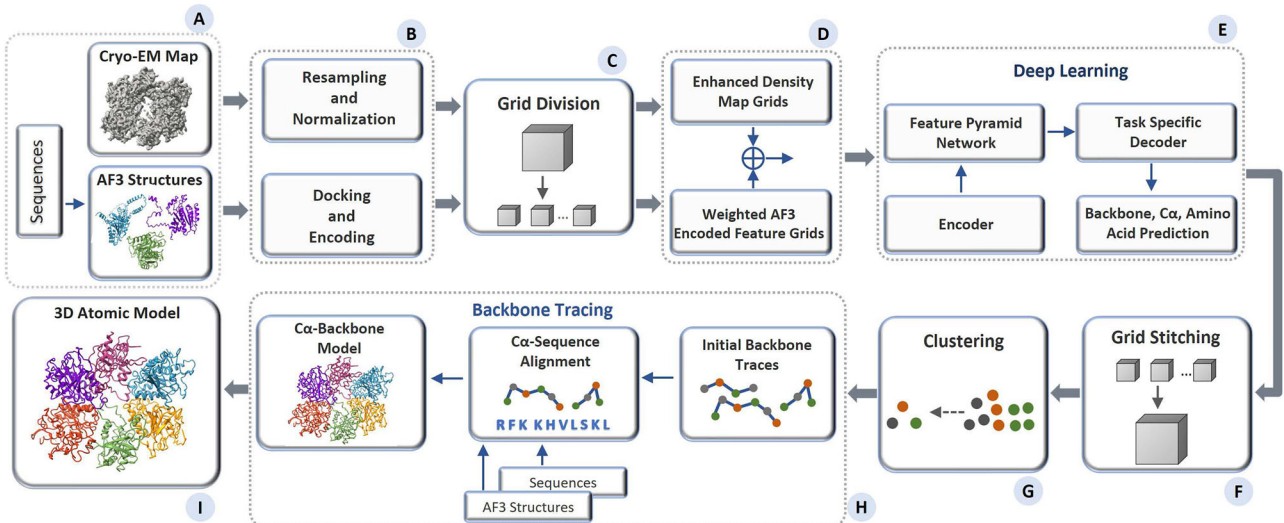

**Fig. 1 | Overview of the atomic structure modeling pipeline of MICA. A** The process begins with multimodal input consisting of a cryo-EM map of a protein, AF3-predicted structures of the individual chains of the protein, and their sequences. **B** The input density map undergoes resampling of each voxel to 1 Å resolution, followed by normalization. The AF3 structure is aligned with the density map through docking, and the features encoding atoms and amino acids in the AF3 structure are extracted. **C** Both the density map and the docked AF3 structure are divided into uniform grids of size 64 × 64 × 64. **D** These cryo-EM and structure grids are processed separately for feature enhancement and then fused together

before passing to a deep learning network. **E** The deep learning network adopts an encoder-decoder based architecture. It utilizes an FPN to perform multi-scale fusion of the features generated by the encoder. The fused features are used by the decoders to predict backbone atoms, Cα atoms, and amino acid types. **F** The prediction results are the grids of size 64 × 64 × 64 and are stitched back to match the shape of the original input density map. **G** The Cα predictions are clustered and combined with amino acid type predictions as input for the backbone tracing. **H** The backbone tracing procedure of EModelX(+AF) is used to build a backbone structure for the protein. **I** The backbone structure is refined to produce a final 3D atomic model.

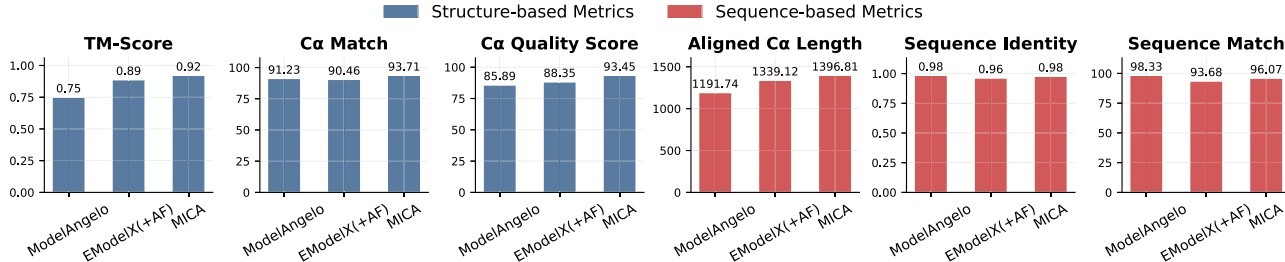

**Fig. 2 | Average performance on Cryo2StructData test dataset for ModelAngelo, EModelX(+AF), and MICA.** MICA achieves superior performance in key evaluation metrics, particularly excelling in TM-score (0.92), Cα match (93.71), and aligned Cα length (1396.31), demonstrating its effectiveness in atomic modeling.

representations. The output of the encoder is used by an FPN to generate multi-scale feature maps, where each map contains distinct levels of spatial detail and semantic information. The feature maps are used by three task-specific decoder blocks with dedicated heads for predicting backbone atoms, Cα atoms, and amino acid types, respectively. The decoders are organized in a hierarchical structure, where the backbone atom decoder uses FPN feature maps as input, the Cα decoder uses both FPN features and the backbone decoder predictions as input, and the amino acid decoder uses FPN features and the predictions of both the backbone atom and Cα decoders as input. The detailed architecture and implementation of the deep learning framework are described in the materials and methods section.

The predicted Cα atoms and amino acid types are used by the backbone tracing procedure adapted from EModelX(+AF)[24] to build an initial backbone model with Cα atoms linked as protein chains and their amino acid types registered. The unmodeled gaps in the initial backbone model are filled by the sequence-guided Cα extension, leveraging the structural information in the AF3-predicted structures for the gaps. This Cα backbone model is transformed to a full-atom model using PULCHRA[27] and further refined against the density maps using phenix.real_space_refine tool[28].

### MICA outperforms ModelAngelo and EModelX(+AF) on the Cryo2StructData test dataset

We compared the performance of MICA with ModelAngelo and EModelX(+AF) on the Cryo2StructData[8] test dataset. The resolution of the density maps in the Cryo2StructData test dataset ranges from 2.05 Å to 3.9 Å with an average of 2.81 Å, and the number of residues per protein varies between 384 and 4128. The sequences of the proteins in the test dataset have <= 25% identity with the ones in the training dataset used to train MICA.

The atomic models built by the three methods from density maps were compared with the corresponding ground truth structures from the Protein Data Bank (PDB)[29]. Six metrics were employed: TM-score, Cα match, Cα quality score, aligned Cα length, sequence identity, and sequence match. These metrics were used to assess the accuracy of structural models from different perspectives. Details on how these metrics were calculated are provided in the materials and methods section.

The average performance evaluation on these density maps for ModelAngelo, EModelX(+AF), and MICA is shown in Fig. 2, while the detailed performance on individual density maps is reported in Supplementary Table S1.

As shown in Fig. 2, MICA outperforms EModelX(+AF) in terms of all six metrics. It also performs better than ModelAngelo in terms of four metrics (TM-score, Cα match, Cα quality score, and aligned Cα length), equally in terms of one metric (sequence identity), and worse in terms of only one metric (sequence match). Specifically, MICA attained a high TM-score of 0.92, which is 22.7% higher than ModelAngelo and 3.4% higher than EModelX(+AF). It achieved a Cα match of 93.71%, 2.7% percentage point higher than ModelAngelo and 3.6% percentage point higher than EModelX(+AF). The Cα quality score of MICA is 93.45, 8.8% higher than ModelAngelo and 5.8% higher than EModelX(+AF).

It is worth noting that among the three structural comparison metrics (TM-score, Cα match, and Cα quality score), TM-score is the most stringent one because it considers both structural match and sequence match between structural models and ground truth structures. According to the threshold of 0.9 for high-accuracy models, MICA is the only method among the three that can build high-accuracy structures from cryo-EM density maps on average. To evaluate the statistical significance of performance differences, we conducted paired t-tests comparing TM-scores between MICA and the two other methods. The analysis revealed that MICA achieved significantly higher TM-scores than both ModelAngelo ($t = 6.85$, $p < 0.001$) and EModelX(+AF) ($t = 3.21$, $p = 0.002$), demonstrating statistically significant improvements over these methods.

Among the three sequence metrics (aligned Cα length, sequence identity, and sequence match) based on structure alignment, aligned Cα length measures the number of residues whose positions can be aligned with ground-truth structures and are therefore considered correctly modeled (i.e., coverage of modeling), while sequence identity quantifies the percentage of identical residues in the alignment between structural model and ground-truth structure (i.e., precision of modeling); Sequence match is similar to sequence identity but it measures the percentage of identical residues in aligned Cα atoms rather than structure alignment. In terms of aligned length, MICA produced an average aligned length of 1398.81, which is 17.2% longer than ModelAngelo and 4.3% longer than EModelX(+AF), indicating it builds more complete models. MICA also achieved the highest sequence identity of 0.98, tied with ModelAngelo's and 2.1% higher than EModelX(+AF)'s 0.96. ModelAngelo achieved the highest sequence match of 98.33%, a few percentage points higher than MICA's 96.07% and EModelX(+AF)'s 93.68%. The higher sequence match of ModelAngelo is due to its structural models being generally much less complete than those of MICA and EModelX(+AF), as indicated by its much shorter aligned Cα length. Since sequence match is the percentage of identical residues within aligned Cα atoms (# identical residues/aligned Cα length), ModelAngelo's fewer aligned Cα atoms create a smaller denominator, leading to higher sequence match scores despite the less complete structural coverage. Considering both aligned Cα length (coverage) and sequence match/identity (precision) together, MICA outperforms ModelAngelo substantially.

The per-map comparison between MICA and ModelAngelo/EModelX(+AF) in terms of each metric is shown in the plots in Fig. 3. These plots demonstrate that MICA performs better than ModelAngelo/EModelX(+AF) on most density maps across most metrics.

### MICA outperforms ModelAngelo and EModelX(+AF) on a standard test dataset

We further compared MICA with ModelAngelo and EModelX(+AF) on the standard test dataset used to evaluate them before[20,24]. In this dataset, the resolution of density maps varies from 1.52 Å to 3.99 Å, with an average resolution of 3.17 Å, while the number of residues per protein spans from 376 to 11,109. The sequences of the proteins in the test dataset have ≤25% identity with the ones in the training dataset used to train MICA. A summary of the average performance of the three methods is presented in Fig. 4,

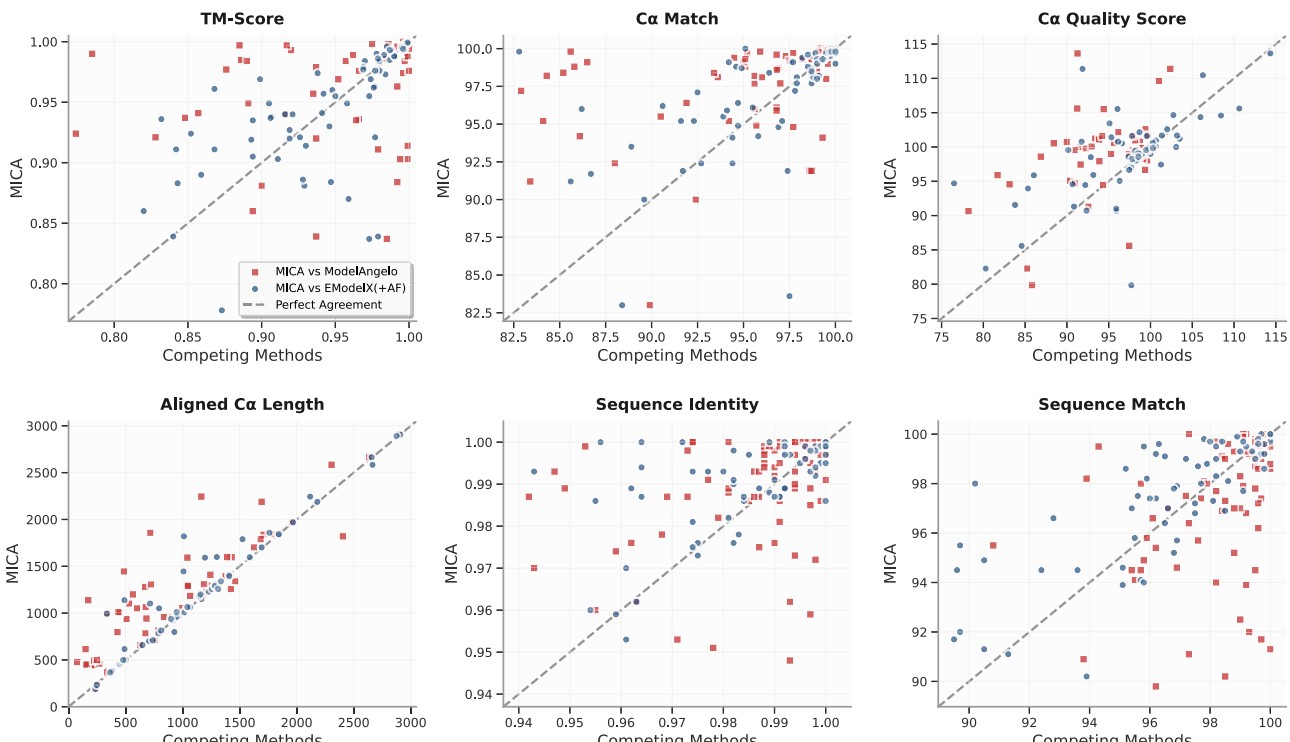

**Fig. 3 | Plots comparing the performance metrics of MICA with those of ModelAngelo and EModelX(+AF) for each cryo-EM density map in the Cryo2StructData test dataset.** Each red square illustrates the comparisons between the score of MICA (y-axis) and that of ModelAngelo (x-axis) for each density map, while the blue circles represent the comparisons between MICA and EModelX(+AF).

Squares and circles positioned above the 45-degree dotted line indicate that MICA outperforms the corresponding method in that metric for cryo-EM maps denoted by the squares and circles. This visualization highlights that MICA performs better on most density maps in terms of most metrics.

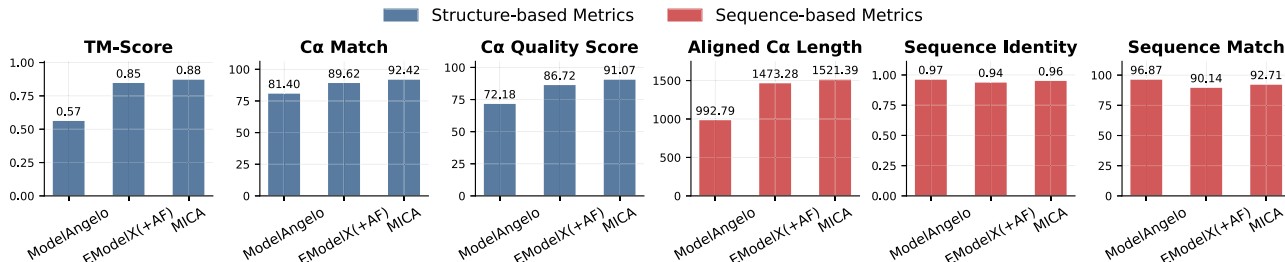

**Fig. 4 | Average performance of MICA in comparison with ModelAngelo and EModelX(+AF) in terms of six metrics on the standard test dataset.** MICA outperforms other methods on key metrics, showing excellent results in TM-score (0.88), Cα match (92.42), and aligned Cα length (1521.39), highlighting its strength in atomic modeling.

while the detailed performance on individual density maps is provided in Supplementary Table S2.

The overall performance of the three methods on the standard test dataset (Fig. 4) is largely consistent with that on the Cryo2StructData test dataset (Fig. 2). Compared to ModelAngelo, MICA demonstrates substantial improvements in structure-based metrics, with TM-score 54.4% higher (0.88 vs 0.57), Cα match 13.5% higher (92.42 vs 81.40), and Cα quality score 26.2% higher (91.07 vs 72.18). Moreover, it generated much more complete models than ModelAngelo, with an average aligned length 53.2% higher (1521.39 vs 992.79), even though it has a 1.0% lower sequence identity (0.96 vs 0.97) and a 4.3% lower sequence match (92.71 vs 96.87). Considering both aligned length (coverage) and sequence identity/match (precision), MICA outperforms ModelAngelo substantially.

Compared to EModelX(+AF), MICA shows a more modest improvement in terms of all the metrics, with gains ranging from 2.1% to 5.0%: TM-score 3.5% higher (0.88 vs 0.85), Cα match 3.1% higher (92.42 vs 89.62), Cα quality score 5.0% higher (91.07 vs 86.72), aligned length 3.3%

higher (1521.39 vs 1473.28), sequence identity 2.1% higher (0.96 vs 0.94), and sequence match 2.9% higher (92.71 vs 90.14). The performance on individual density maps for all metrics is shown in the plots in Fig. 5. These plots highlight the improved performance of MICA across most density maps in terms of most metrics.

In terms of the most stringent TM-score metric, according to the paired t-test, the performance of MICA is significantly better than ModelAngelo ($t = 12.93$, $p < 0.0001$) and EModelX(+AF) ($t = 3.80$, $p = 0.0003$).

Figure 6 shows four examples where MICA outperformed ModelAngelo and EModelX(+AF) (Fig. 6A–D), two examples where it performed slightly better (Fig. 6E, F), and two examples where it performed worse than one or two methods (Fig. 6G, H). For density map EMD-15685 (Fig. 6A), with 1156 residues, MICA obviously modeled the helical residues in the bottom-right low-resolution region of the density map better than the other two methods. In the case of EMD-26595 (Fig. 6B), with 2400 residues, it modeled the overall fold of the protein better than the other two methods. For EMD-26841 (Fig. 6C) with 4834 residues and EMD-33528 (Fig. 6D),

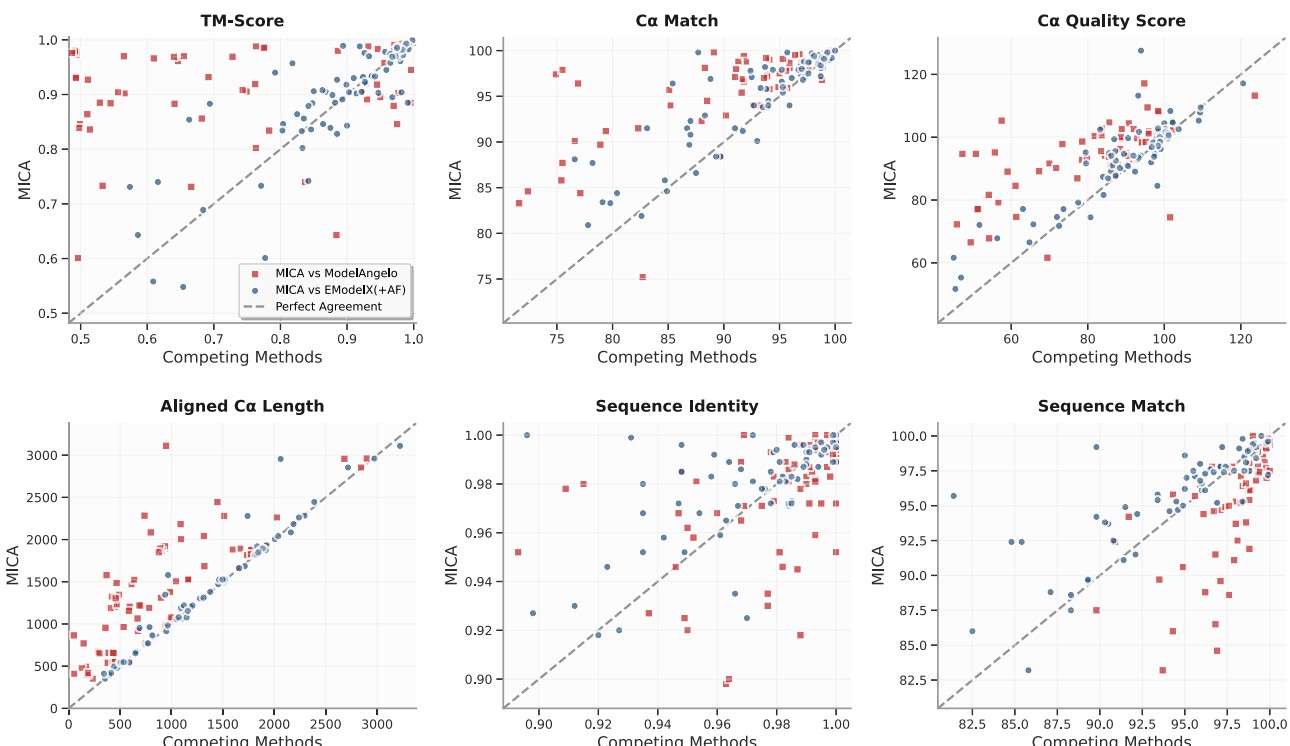

**Fig. 5 | Comparison of the performance of MICA with that of ModelAngelo and EModelX(+AF) for each cryo-EM density map on the standard test dataset.** For each map, the score of MICA is plotted against that of either ModelAngelo (red squares) or EModelX(+AF) (blue circles).

with 4770 residues, MICA has a higher TM-score compared to ModelAngelo and EModelX(+AF), highlighting its advantage in modeling very large protein assemblies. Similarly, for the challenging large density maps EMD-14869 (Fig. 6E) with 11,109 residues and EMD-28866 (Fig. 6F) with 5553 residues, MICA's atomic model has substantially higher TM-score than ModelAngelo's and slightly higher than EModelX(+AF)'s. For EMD-26974 and EMD-14847 (Fig. 6G, H), even though the models built by MICA are of good quality, they are moderately less accurate than the models built by ModelAngelo and/or EModelX(+AF).

## MICA demonstrates robust performance across protein sizes and cryo-EM map resolutions

We investigated how protein length (total number of residues in a protein) influences the modeling performance of MICA in terms of TM-score, Cα match and aligned Cα length in comparison with ModelAngelo and EModelX(+AF) (Fig. 7). Linear regression analysis shows that the aligned length exhibits the strongest correlation with sequence length, with MICA demonstrating higher predictability ($R^2 = 0.84$, slope = 0.78) compared to EModelX(+AF) ($R^2 = 0.75$, slope = 0.73) and ModelAngelo ($R^2 = 0.51$, slope = 0.61), indicating MICA's modeling capability scales well with protein size. TM-score and Cα match have weak correlations with protein lengths for all three methods, indicating that their modeling performance is rather stable and not sensitive to protein length. Moreover, the regression line of MICA lies above those of the other two methods, indicating that it generally produces models of better quality across protein lengths.

We also assessed how the performance of MICA varies with respect to cryo-EM density map resolution in comparison with ModelAngelo and EModelX(+AF) in terms of TM-score, Cα match, and aligned Cα length (Fig. 8). Linear regression analysis reveals that these metrics degrade as map resolution worsens for all methods to a different degree. TM-score analysis demonstrates that MICA has the smallest negative slope (slope = −0.07, $R^2 = 0.04$) compared to ModelAngelo (slope = −0.30, $R^2 = 0.20$) and EModelX(+AF) (slope = −0.08, $R^2 = 0.04$), indicating a small reduction in TM-score for MICA and EModelX(+AF) and a substantial reduction for

ModelAngelo as map resolution deteriorates. The low $R^2$ values for MICA and EModelX(+AF) suggest their performance is much less sensitive to map resolution. Cα match performance follows a similar pattern, with MICA exhibiting the smallest negative slope (slope = −5.56, $R^2 = 0.05$) compared to ModelAngelo (slope = −25.10, $R^2 = 0.35$) and EModelX(+AF) (slope = −6.90, $R^2 = 0.04$). ModelAngelo's higher $R^2$ value indicates greater resolution dependence, while MICA's much lower $R^2$ reflects more consistent performance across varying resolutions. Aligned Cα length also shows MICA's superior resilience (slope = −199.64, $R^2 = 0.01$) compared to ModelAngelo (slope = −580.56, $R^2 = 0.09$) and EModelX(+AF) (slope = −206.07, $R^2 = 0.01$). These results collectively demonstrate MICA is most robust across all evaluated metrics, with minimal quality degradation as map resolution worsens. The consistently low $R^2$ values for MICA indicate that its performance is largely resolution-independent, establishing it as a reliable modeling method for cryo-EM maps of resolution 1–4 Å.

## MICA builds high-accuracy protein structures for new cryo-EM density maps released after January 1, 2025

To further assess how well MICA can generalize to newly determined cryo-EM density maps, we applied it to new cryo-EM density maps in the test_2025 dataset released after January 1, 2025 (much later than the maps used to train MICA). The 12 new cryo-EM density maps in test_2025, with resolutions varying from 2.08 Å to 3.82 Å and an average resolution of 2.96 Å, were used by MICA to build structural models, which were compared with the corresponding ground truth structures released in the PDB.

The results are presented in Supplementary Table S3. MICA achieved high modeling accuracy on the new density maps, which is even higher than or comparable to the accuracy attained on the Cryo2StructData test dataset and the standard test dataset. For instance, the average TM-score is 0.93, while the average of the other metrics (Cα match, Cα quality score, sequence identity, and sequence match) is all above 0.96, indicating high-accuracy structural models were constructed for these density maps. The structural models built for 9 out of 12 cryo-EM density maps are of high accuracy (TM-score > 0.9). Moreover, the aligned Cα length and the model length

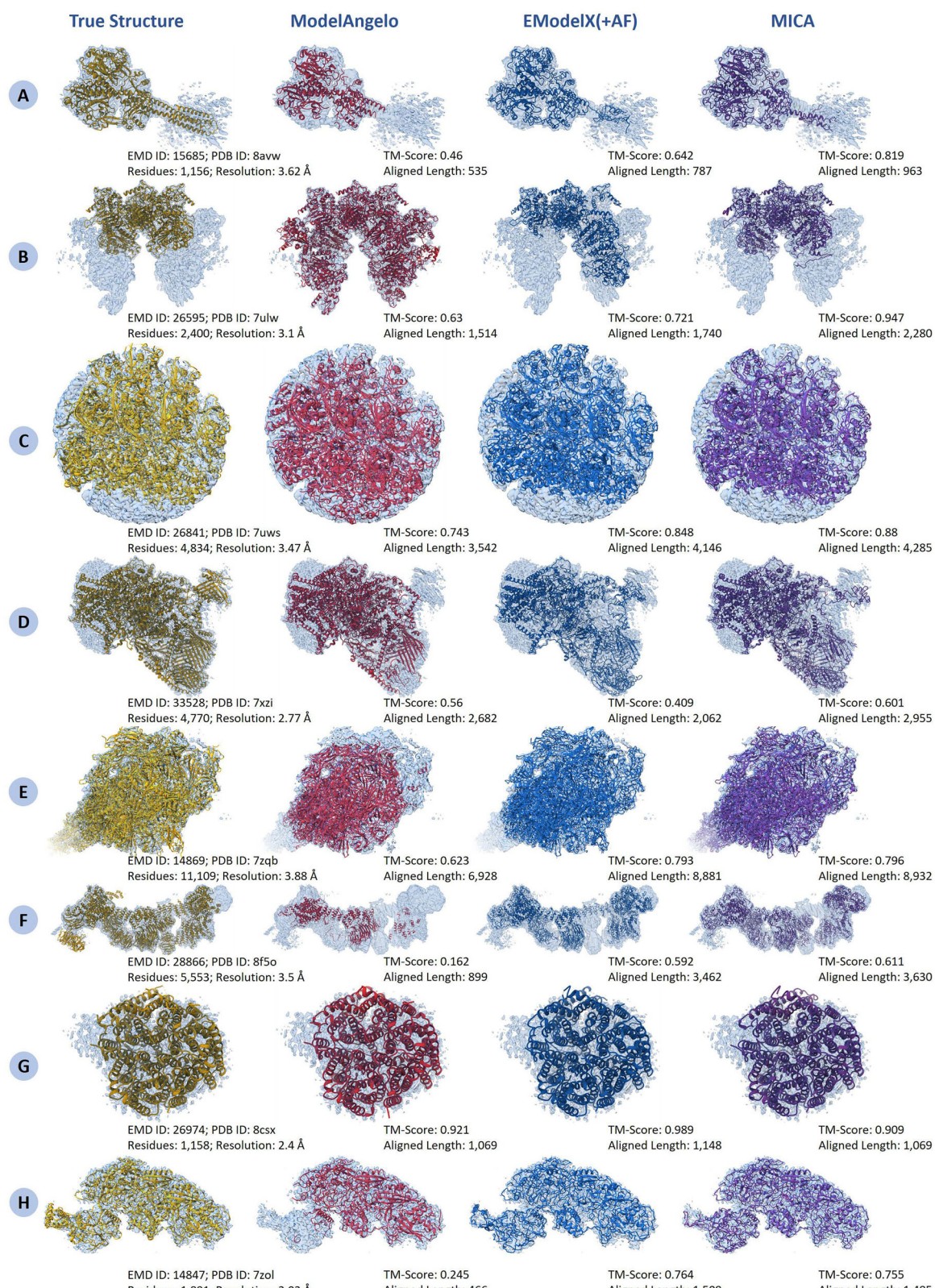

**Fig. 6 | Comparison of structural models generated by ModelAngelo, EMo-delX(+AF), and MICA against reference structures superimposed on their corresponding cryo-EM density maps across eight diverse protein complexes.** **A** EMD ID: 15,685 (PDB ID: 8avw; 1156 residues; 3.62 Å resolution). **B** EMD ID: 26,595 (PDB ID: 7ulw; 2400 residues; 3.1 Å resolution). **C** EMD ID: 26841 (PDB ID: 7uws; 4834 residues; 3.47 Å resolution). **D** EMD ID: 33528 (PDB ID: 7xzi; 4700 residues; 2.77 Å resolution). **E** EMD ID: 14,869 (PDB ID: 7zqb; 11,109 residues; 3.88 Å resolution). **F** EMD ID: 28,866 (PDB ID: 8f5o; 5533 residues; 3.5 Å resolution). **G** EMD ID: 26,974 (PDB ID: 8csx; 1158 residues; 2.4 Å resolution). **H** EMD ID: 14,847 (PDB ID: 7zol; 1891 residues; 3.03 Å resolution).

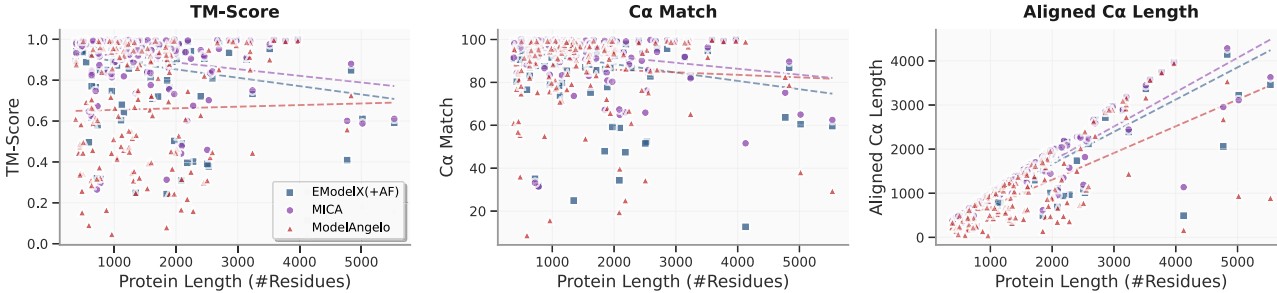

**Fig. 7 | Impact of protein length on the modeling performance of three methods using both the Cryo2StructData and standard test datasets.** The TM-score, Cα match, and aligned Cα length of the structural models built by three methods (ModelAngelo represented by red triangles, EModelX(+AF) represented by blue squares, and MICA represented by purple circles) are plotted against protein lengths, respectively. The regression lines for these data points are depicted, with the red dotted line representing ModelAngelo, the blue dotted line representing

EModelX(+AF), and the purple dotted line representing MICA. The correlation between TM-score or Cα match and protein length are weak, indicating the three methods are not sensitive to protein length. The aligned Cα length has a high correlation with protein length, as expected, because more residues can be modeled for larger proteins. The regression line (purple) of MICA lies above the other two methods, indicating that it generally builds models of better quality across protein length.

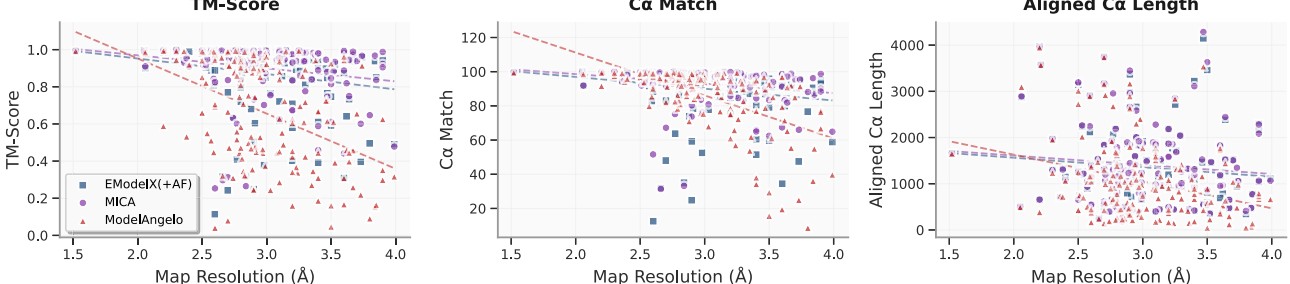

**Fig. 8 | Impact of cryo-EM map resolution on modeling performance of three methods on both the Cryo2StructData and standard test datasets.** The TM-score, Cα match, and aligned Cα length of the structural models built by three methods (ModelAngelo represented by red triangles, EModelX(+AF) represented by blue squares, and MICA represented by purple circles) are plotted against map resolution, respectively. The regression lines for these data points are depicted, with the red dotted line representing ModelAngelo, the blue dotted line representing

EModelX(+AF), and the purple dotted line representing MICA. Linear regression analysis reveals that all three metrics degrade as map resolution worsens across all methods to some degree, with negative slopes indicating declining performance. However, MICA demonstrates the smallest negative slopes for TM-score, Cα match, and aligned Cα length, indicating minimal quality reduction compared to ModelAngelo and EModelX(+AF) as resolution deteriorates, demonstrating its improved performance.

(the number of residues of a structural model) are very close to the true protein length, showing that the models built by MICA are rather complete. The results demonstrate that MICA can be applied to automatically build high-accuracy protein structures from high-resolution cryo-EM density maps in real-world applications (see the structural models built for all 12 density maps in Fig. 9).

### Impact of AF3 model quality on MICA's performance

To assess MICA's robustness under varying input model quality conditions, we investigated how the quality of AF3 structural predictions affects the final structure modeling performance. This analysis is crucial for understanding MICA's dependence on AF3 features and its practical applicability when AF3 predictions vary in quality.

We categorized all 160 density maps from both the Cryo2StructData test dataset and Standard test dataset into three groups based on the quality of their corresponding AF3 models as measured by TM-score: low quality AF3 models (TM-score < 0.5, $n = 39$ maps), medium quality AF3 models ($0.5 \leq$ TM-score < 0.8, $n = 38$ maps), and high quality AF3 models (TM-score $\geq 0.8$, $n = 83$ maps). For each category, we calculated the average TM-scores of both the input AF3 models and the structural models built by MICA to examine the correlation between input quality and final performance. The results, presented in Supplementary Fig. S1, demonstrate that while MICA's performance positively correlates with AF3 quality, the method maintains robust modeling capabilities across all AF3 quality categories. For the density maps with low-quality AF3 models (average TM-score of 0.32), MICA builds significantly better final structural models with

an average TM-score of 0.77, representing a 2.4-fold improvement over the input AF3 models. This substantial enhancement indicates MICA's ability to leverage cryo-EM density information effectively even when AF3 structural predictions are unreliable. With medium-quality AF3 models (average TM-score of 0.62) as input, the average TM-score of MICA-built structural models increases to 0.89, while for high-quality AF3 models (average TM-score of 0.95), MICA achieves near-optimal MICA performance of 0.96 TM-score.

This improvement reflects MICA's intelligent feature integration architecture, where the feature gate module automatically suppresses unreliable AF3 predictions while amplifying high-confidence information. The random feature masking strategy during training (40% cases using cryo-EM only, 60% cases using combined features) prevents over-dependence on AF3 models, ensuring robust performance regardless of AF3 prediction quality. The consistent improvements across all categories also confirm that MICA treats cryo-EM density as its primary information source while utilizing AF3 features as supplementary information. Even with low-quality AF3 input, MICA produces high-quality models by leveraging cryo-EM density information, demonstrating its broad applicability when AF3 prediction quality varies significantly.

### Impact of domain docking on MICA's performance

To evaluate how domain docking accuracy affects MICA's performance, we analyzed the results on the 160 test density maps from both Cryo2S-tructData and Standard test datasets based on docking success. We

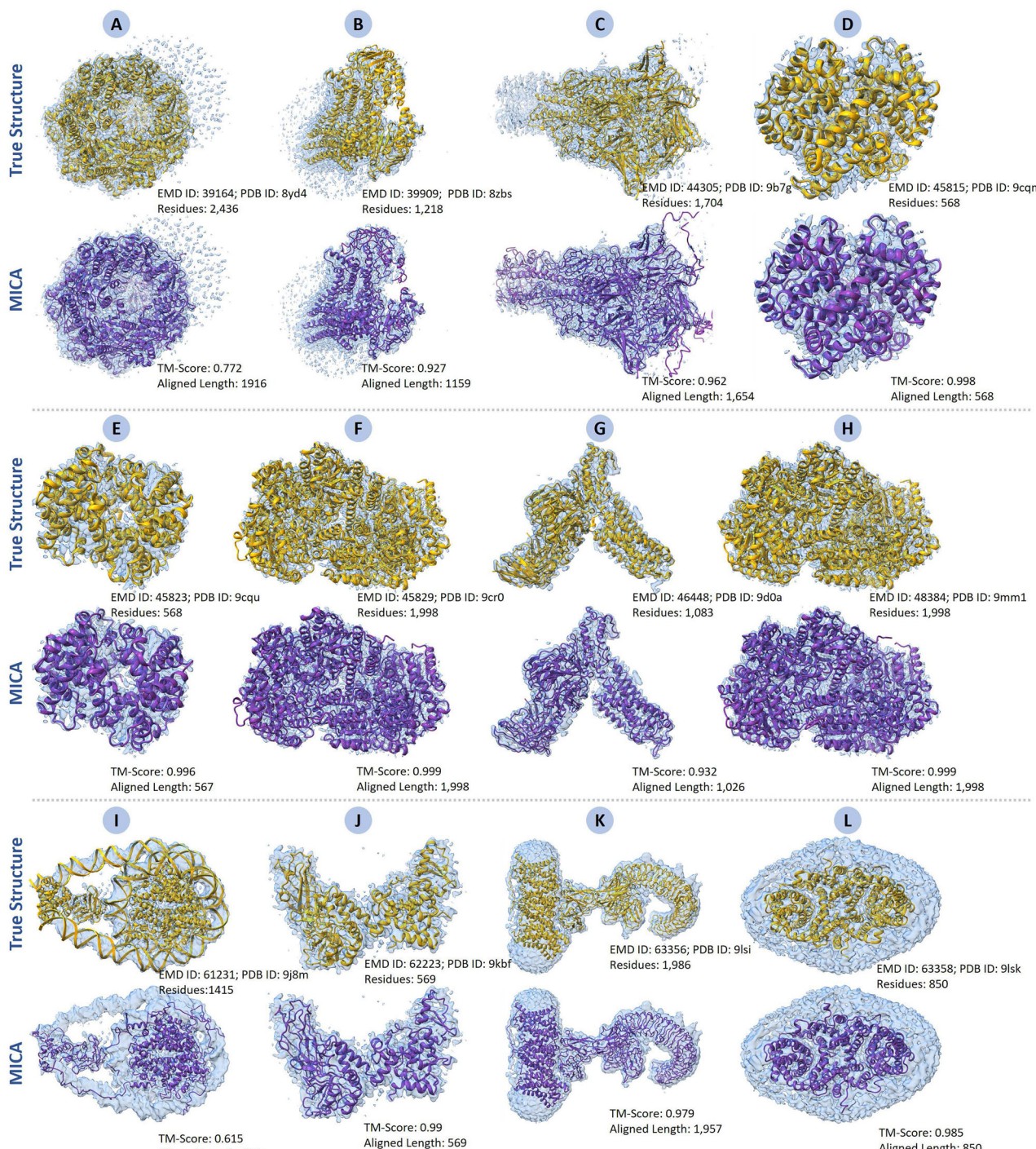

**Fig. 9 | Structural models generated by MICA for 12 new cryo-EM density maps released after January 1, 2025. A** EMD ID: 39,164 (PDB ID: 8yd4; 2436 residues; 3.69 Å resolution). **B** EMD ID: 39,909 (PDB ID: 8zbs; 1218 residues; 2.96 Å resolution). **C** EMD ID: 44,305 (PDB ID: 9b7g; 1704 residues; 2.61 Å resolution). **D** EMD ID: 45,815 (PDB ID: 9cqm; 568 residues; 2.55 Å resolution). **E** EMD ID: 45,823 (PDB ID: 9cqu; 568 residues; 2.72 Å resolution). **F** EMD ID: 45,829 (PDB ID: 9cr0; 1998 residues; 2.08 Å resolution). **G** EMD ID: 46448 (PDB ID: 9d0a; 1083 residues; 3.1 Å resolution). **H** EMD ID: 48384 (PDB ID: 9mm1; 1998 residues; 2.08 Å resolution). **I** EMD ID: 61,231 (PDB ID: 9j8m; 1415 residues; 3.82 Å resolution). **J** EMD ID: 62,223 (PDB ID: 9kbf; 569 residues; 3.74 Å resolution). **K** EMD ID: 63,356 (PDB ID: 9lsi; 1986 residues; 3.3 Å resolution). **L** EMD ID: 63,358 (PDB ID: 9lsk; 858 residues; 2.9 Å resolution).

categorized cases into fully docked (*n* = 150 density maps where all AF3 model domains successfully docked) and partially docked (*n* = 30 density maps with incomplete domain docking).

For fully docked cases, AF3 input models have an average TM-score of 0.74 compared to MICA's 0.91, while in partially docked cases AF3 input models have an average TM-score of 0.64 vs MICA's 0.86 (Supplementary Fig. S2). The reduced accuracy in both AF3 and MICA models for partially

docked cases demonstrates some correlation between domain docking success and structural prediction quality.

However, even when AF3 domains are only partially docked, MICA still achieves a high average TM-score of 0.86, only 0.05 less than the TM-score of 0.91 when all the domains are docked successfully. This can be largely attributed to MICA's domain-wise quality control mechanism. Even when the overall AF3 model is inaccurate, some individual domains that are

well predicted can still dock successfully and contribute valuable features to enhance prediction accuracy. Conversely, poorly predicted domains are automatically filtered out through docking failure, preventing contamination from erroneous structural information. Moreover, in partially docked cases, MICA compensates for missing AF3 features by relying more heavily on its cryo-EM-based features, though some performance reduction may also stem from inherently challenging density regions with lower resolution or increased structural complexity.

This result confirms MICA's intelligent feature integration strategy, where the architecture selectively utilizes reliable AF3 domains while maintaining robust performance through cryo-EM-driven modeling when structural predictions are incomplete or unreliable. During training, this selective incorporation of both correct and incorrect docking results serves as implicit data augmentation, encouraging the learning of robust feature representations rather than simple memorization of AF3 structural information.

### MICA's performance on cryo-EM density maps with lower (4–6 Å) resolution

While MICA is primarily designed and trained for high-resolution density maps in the 1–4 Å range, we evaluated its capability beyond this optimal resolution range by testing it on 8 representative density maps with resolutions of 4–6 Å. The results presented in Supplementary Table S4 reveal variable performance that reflects the inherent challenges of lower-resolution structural modeling. MICA achieved good performance (TM-score > 0.7) for 4 out of 8 cases, including EMD-62354 (TM-score: 0.789) and EMD-49996 (TM-score: 0.897), while showing suboptimal results for more challenging cases such as EMD-43679 (TM-score: 0.268) and EMD-51169 (TM-score: 0.484). This variable performance is expected, as reduced density details at lower resolutions make it increasingly difficult to distinguish individual atoms and side chain orientations that are critical for accurate atomic modeling. While MICA maintains some modeling capability beyond 4 Å resolution, users should be careful when applying it to maps with significantly lower resolution.

However, MICA demonstrated superior robustness on the density maps of 4-6 Å compared to ModelAngelo and EModelX(+AF), which either failed to generate complete models or produced only fragmented models for them (Supplementary Table S4), highlighting MICA's comparative advantage even when operating beyond its optimal resolution range.

### Typical execution time

To assess the computational efficiency of the automated MICA pipeline, we benchmarked its execution time across 18 protein structures ranging from 376 to 3238 residues (Supplementary Table S5). Total processing time ranged from 10.63 minutes for small proteins (495 residues, 3 domains) to 303.53 minutes for very large proteins (2859 residues, 12 domains). The domain docking step, performed through Phenix using CPU-based computation, consistently represented the primary computational bottleneck, accounting for 85–97% of total execution time. Notably, execution time scaled non-linearly with residue count due to varied complexity in inter-domain and inter-chain interactions. Based on the data in Supplementary Table S5, proteins with 300–700 residues typically require 10–30 min of execution time, proteins with 700–1400 residues need 30–90 min for processing, proteins with 1400–2500 residues require 150–240 min, and proteins with more than 2500 residues demand 240–300+ min. All benchmarks were performed using 24 CPU cores and one NVIDIA A100 GPU with 80GB of memory.

### Discussion

We developed MICA, a fully automated deep learning method for generating high-accuracy protein structural models from high-resolution cryo-EM density maps of 1–4 Å in conjunction with AF3-predicted structures. A major innovation in MICA is that it integrates density maps and AF3-predicted structures to generate more informative input features to predict

backbone structures of proteins for the first time, which is different from the other deep learning methods that only combine structural models built from density maps with AF3-predicted structures at the output level in the post-processing step. MICA is able to build significantly more accurate and more complete structural models from cryo-EM density maps than both ModelAngelo and EModelX(+AF). The improvement is primarily due to the multimodal integration of cryo-EM data and AF3-predicted structure at the input level by a novel encoder-decoder model based on the FPN. The multiscale input processing of cryo-EM maps fused with AF3-based structural features creates enriched feature representations for predicting the backbone structure of proteins.

Moreover, the deep learning architecture of MICA is different from traditional CNNs and transformers used in the field, it is optimized specifically for cryo-EM data and enables highly accurate predictions of backbone atoms, Cα positions, and amino acid types, resulting in more accurate structural models.

The results show that MICA achieved outstanding modeling performance in terms of multiple complementary metrics such as TM-score, Cα match, aligned Cα length, and Cα quality score on multiple datasets. Particularly, it generalizes well to the newly released cryo-EM density maps, indicating it is ready to be used for automatically building high-accuracy protein structural models from cryo-EM density maps and AF3-predicted structures in real-world settings, releasing human experts from laborious manual intervention of the cryo-EM modeling process.

The increase in Cα match, TM-score, Cα quality score, and aligned Cα length for MICA compared to ModelAngelo and EModelX(+AF) is primarily attributed to its ability to predict a higher number of Cα atoms accurately, demonstrating the effectiveness of the multimodal data integration and robust deep learning architecture for backbone atom and Cα atom prediction. Additionally, the improvement in Cα match and TM-score from 92.42% and 0.88 on the standard dataset with an average map resolution of 3.17 Å to 93.71% and 0.92 on the Cryo2StructData dataset with an average resolution of 2.81 Å underscores the importance of higher resolution of density maps for accurate structure prediction. As the cryo-EM community continues to generate increasingly higher-resolution density maps, MICA's performance is expected to improve automatically on future datasets, as indicated by its excellent performance on the newly released high-resolution cryo-EM density maps, benefiting from enhanced structural details and reduced noise in experimental density maps.

Although MICA is able to build high-accuracy models from most high-resolution cryo-EM density maps, it still encounters difficulties in constructing highly accurate models for some large protein complexes, where the performance may be compromised by challenges in sequence and chain registration, particularly when structural information in the cryo-EM maps is insufficient due to missing or noisy density values or when AF3-predicted structures align poorly with experimental density maps due to low map resolution in some regions or errors in the predicted structures. Further improving modeling accuracy requires not only accurate identification of Cα atoms, but also correct linkage between Cα atoms, amino acid types, and sequence and chain registration across multiple chains.

In the future, we plan to improve the sequence and chain registration accuracy of MICA to further enhance the accuracy and completeness of atomic models. We will also explore the symmetry of multiple chains in protein complexes when mapping amino acid sequences to Cα atoms. Additionally, we aim to incorporate advanced side-chain prediction algorithms into the deep learning framework to further enhance all-atom modeling rather than adding side chains in the last post-processing step.

Another important enhancement involves developing local resolution integration capabilities through a multi-faceted approach. This framework will combine resolution-aware feature weighting, where local resolution estimates are provided as an additional input to dynamically modulate cryo-EM feature importance, with confidence-calibrated scoring that generates per-residue uncertainty estimates based on local resolution quality metrics and geometric consistency between experimental and computational features. The system will also integrate resolution-guided AF3 integration that

progressively increases structural prior weighting in lower-resolution regions to ensure physically reasonable predictions when experimental constraints are weak. This integrated approach will enable MICA to maintain high precision in well-resolved regions while providing robust, confidence-quantified alternative models in challenging areas, ultimately delivering more reliable and interpretable structural models across heterogeneous cryo-EM datasets.

## Materials and methods
### Datasets
From the Cryo2StructData dataset[8], we collected 550 density maps released by April 2023 that exhibit high correlation with the true structures deposited in the PDB[29] to train and validate MICA, with 80% allocated for training and the remaining 20% for validation. The complete information of the training/validation dataset is available in Supplementary Table S6.

The performance of MICA was compared with the widely used ModelAngelo and the recently introduced EModelX(+AF) on two independent test datasets. The first test dataset was taken from Cryo2StructData[8], and the second is a standard test dataset used to evaluate ModelAngelo[20] and EModelX(+AF)[24] before. None of the density maps in the two datasets was included in the training or validation data of MICA. Furthermore, the sequences of the protein chains in the two datasets were compared with the training data of the three methods using MMSeqs2[30] to remove any proteins that have greater than 25% sequence identity with any chain used in training. This filtering resulted in 80 density maps left in each dataset to compare the three methods. The details of the two test datasets are presented in Supplementary Table S7 and Supplementary Table S8.

Finally, we randomly selected 12 cryo-EM density maps released after January 1st, 2025, from the Electron Microscopy Data Bank (EMDB)[31] to create the third test dataset (called test_2025) to evaluate how well MICA generalizes to new density maps in the real-world setting.

AF3[25] was used to predict the structure for each chain of the protein of each cryo-EM density map from its sequence. Both the density maps and the AF3-predicted structures are used as input for MICA to build the final structures for proteins.

The ground truth protein structure for each cryo-EM density map in all the datasets above was retrieved from the PDB to train, validate, and test MICA.

### Input preprocessing and label preparation
Because cryo-EM density maps have varying voxel spacings and voxel values, it is necessary to standardize them before using them with deep learning models. To address this, we resampled each density map with a constant voxel spacing of 1 Å, ensuring consistency across all the maps. After resampling, the maps were normalized using median-based background subtraction, followed by 99.9th percentile clipping to reduce the influence of outliers and the min-max normalization to scale values between 0 and 1. This approach is robust to outliers and preserves signal structure while standardizing the dynamic density value range across different cryo-EM density maps. The standardized density values are the set of input features representing the density map modality.

The AF3-predicted structures are aligned with the corresponding standardized density maps to generate another set of input features representing the predicted structure modality as follows. For each cryo-EM density map, AF3 was used to predict structures for each chain from the protein's FASTA sequence. The AF3 server generates 5 different models (ranked 0-4) for each unique chain, from which we selected "model 0" with the highest confidence score. Thus, the total number of atomic models from AF3 predictions corresponds to the number of unique chains in the protein complex. These selected atomic models were then divided into structural domains using Merizo[32] and subsequently docked into the density map using the Phenix.dock_in_map[7] command to align them within the same 3D coordinate system as the cryo-EM data. The docked domain structures of the same chain were combined into a single structure situated in the same 3D grid as the cryo-EM density map. A channel-wise binary encoding was

performed for the backbone Cα atom (Cα), nitrogen atom (N), carbonyl backbone carbon atom (C), backbone oxygen atom (O), and the 20 different amino acids located in each voxel in the 3D grid, creating a 24-channel encoded volume. The shape of the resulting encoding of the AF3-predicted structures matched exactly the dimensions of the corresponding cryo-EM density map.

Three sets of ground truth labels for the backbone atoms, Cα atoms, and amino acid types, respectively, were prepared to train MICA as follows. We first initialized three zero-filled ground truth masks for each normalized density map. We used the protein structure corresponding to the density map to create the ground truth backbone atom mask. Voxels containing any backbone atoms were labeled as "3," while those containing other atoms were labeled as "2." Neighboring voxels adjacent to any atom in the structure were assigned the label "1," and all remaining non-structural voxels were labeled as "0." Similarly, for the Cα atom mask, we labeled voxels containing Cα atoms as "3" and other atoms as "2", neighboring voxels of atoms as "1", and the remaining non-structural voxels as 0. For the amino acid ground truth mask, we labeled the voxels containing Cα atoms and the neighboring voxels with labels "1"–"20" for different types of amino acids of the Cα atoms, and the remaining voxels as "0".

In preparing the ground truth masks above, the location of each voxel is accessed by its index $(i, j, k)$. However, the corresponding true protein structure used for labeling the voxels is in a different 3D coordinate system $(x, y, z)$. Therefore, we calculated the corresponding indices of each atom in the atom ground truth mask from its atomic coordinates using Eq. 1. In this formula, $(i, j, k)$ are the grid indices of the atom in the mask; $(x, y, z)$ are the coordinates from the PDB file; $origin_x$, $origin_y$, $origin_z$ are the origins of the $x$, $y$ and $z$ axes, respectively, found in the normalized density map; and $voxel_x$, $voxel_y$, $voxel_z$ are the voxel sizes of the $x$, $y$ and $z$ axes, respectively, found in the normalized density map.

$$i = \left\lceil \left( \frac{\lfloor (z - origin_z) }{voxel_z} \right) \right\rceil; j = \left\lceil \left( \frac{\lfloor (y - origin_y) }{voxel_y} \right) \right\rceil; k = \left\lceil \left( \frac{\lfloor (x - origin_x) }{voxel_x} \right) \right\rceil$$

(1)

The final preprocessing step involves partitioning cryo-EM density maps into uniform grids using a sliding window approach with contextual padding. Each map was divided into non-overlapping 48³ voxel grids, with 8-voxel padding applied to all boundaries to create 64³ voxel processing windows. Zero-padding was applied to map edges to ensure complete spatial coverage and contextual padding to preserve spatial relationships at grid boundaries, ensuring consistent feature extraction and preventing edge artifacts. The same grid partitioning scheme was applied to the encoding grids for AF3-predicted structures and the ground truth masks to maintain spatial correspondence. This grid partitioning enables efficient GPU memory utilization while preserving spatial context for accurate boundary predictions and ensuring consistent computational workload across all data modalities.

### Deep learning model for integrating cryo-EM density maps and AF3-predicted structures to predict backbone atoms, Cα atoms, and amino acid types
Cryo-EM density maps and AF3-predicted structures provide complementary information about the underlying true structures of proteins. Density maps contain experimental data about the locations of atoms and amino acids in ground-truth structures, but some regions of density maps may be of low resolution or have no density values and therefore do not contain sufficient information for building atomic structures for the regions. In contrast, AF3 can predict the structures of the entire region of proteins. Particularly, it can predict the structures of individual protein domains rather accurately. However, AF3 still cannot accurately predict interactions between individual domains in some multi-domain proteins and the interactions between individual chains in many protein complexes, particularly large ones. Therefore, it is necessary to combine the two sources of

information to improve the accuracy of cryo-EM based protein structure modeling[11].

Recently, two methods have been developed to use AlphaFold-predicted structures to refine incorrectly modeled regions in the protein structural models built from cryo-EM density maps[23,24]. However, this data integration at the output level in the postprocessing step does not fully leverage the structural information in AlphaFold-predicted structures. In this work, MICA adopts a novel approach to use a deep learning model to take both cryo-EM density maps and AF3-predicted structures as input and integrate them together through representation learning to build protein structures in the first place, i.e., to predict backbone atoms, Cα atoms, and amino acid types.

MICA employs a multimodal encoder-decoder learning framework that combines standardized cryo-EM maps with AF3 structures-derived encodings through a dual-processing pathway (Fig. 10A). This pathway learns a representation for each of the two data modalities first before merging them together.

Specifically, 3D grids of cryo-EM density maps with dimensions of $64^3$ and a single feature channel (i.e., density value) are fed to a multi-scale convolution block. Multiple 3D convolutional filters with kernel sizes of 3, 5, 7, and 9 in the block generate multi-scale features that are concatenated to produce a feature volume of $128 \times 64^3$. This feature volume undergoes self-attention processing, which includes average pooling and a couple of convolutional blocks, with the output multiplied by the concatenated features from the previous block to yield final representations called enhanced density map grids (Fig. 10A).

In parallel, another input, the AF3 structure encoded grids of size $64^3$ with 24 channels, undergo feature convolution with a filter size of $3 \times 3 \times 3$ to generate a feature volume of $64 \times 64^3$ in a feature convolution block. The feature volume is further processed through a couple of convolutional blocks called feature gates, whose output is multiplied by the feature convolutional block output to generate final weighted AF3 encoded features.

These weighted AF3 features are then concatenated with the enhanced density map features, which are processed by a fusion convolutional block to generate the final combined representation of both cryo-EM maps and AF3 structures with size $64 \times 64^3$ (Fig. 10A). If AF3 encoding features are unavailable for a cryo-EM density grid, i.e., if no AF3-predicted structure can be aligned with the grid, only the enhanced experimental density map grids are used.

The multimodal representation is then passed through a progressive encoder stack with three encoder blocks featuring increasing feature depth (Fig. 10B). The 64-channel feature volume is transformed using 128, 256, and 512 filters of size $3 \times 3 \times 3$ in three encoder blocks, respectively, to create hierarchical feature representations. Each encoder block consists of a residual dense block, dual attention block, and a transition block. The residual dense block contains three convolutional blocks with filter size $3 \times 3 \times 3$, creating dense connections between layers that allow information to flow directly from earlier to later layers, followed by a squeeze-and-excitation block that emphasizes important feature channels while suppressing less useful ones. The output is then passed to a dual attention block, which calculates both local and global attention. The local attention block uses convolution filters of size $3 \times 3 \times 3$ to capture local spatial relationships, while the global attention block employs average pooling and convolutional blocks to capture global context and channel attention weights. These weights are multiplied by the input, concatenated with local features, and passed to the transition block with a convolutional filter of size $3 \times 3 \times 3$, to generate the encoder output. The encoder outputs are of dimensions $128 \times 64^3$, $256 \times 64^3$, and $512 \times 64^3$, respectively.

The FPN takes three multi-scale feature maps from the three encoders as inputs, each containing different levels of spatial detail and semantic information, to further process them (Fig. 10B). The first module of FPN applies lateral convolution to harmonize features from the three encoders into a uniform 64-channel representation. The upsampling module then upsamples coarser features to match the highest resolution using trilinear interpolation. These interpolated features are smoothed using three

sequential convolution blocks with filter size $3 \times 3 \times 3$. Learnable adaptive weights are applied to each scale, and the smoothed features are concatenated, producing 192 feature channels. The softmax-normalized learnable weighting allows the network to automatically determine the importance of each scale during training.

Overall, the hierarchical architectural design within the three encoders enables learning of increasingly complex multimodal representations of the input data, which are utilized by FPN for multi-scale feature aggregation at different resolution levels before feeding into the final prediction stage.

The final stage of the deep learning model comprises three task-specific decoder blocks with separate heads for predicting backbone atoms, Cα atoms, and amino acid types, respectively, organized in a cascading fashion (Fig. 10B). Each decoder consists of convolutional blocks where feature channels are progressively reduced to match the number of output classes. The backbone atom decoder produces four output channels corresponding to 4 different backbone labels. The Cα decoder takes input from the FPN combined with the output of the backbone atom decoder to produce four output channels corresponding to four different Cα labels. The amino acid decoder takes the FPN output concatenated with the outputs of both the backbone decoder and the Cα decoder to produce 21 output channels (20 different amino acid types and a no amino acid type). This design reflects the natural biological hierarchy where backbone structure determines Cα positions, and both can influence amino acid placement. By feeding backbone atom predictions to the Cα decoder, and both to the amino acid decoder, the model learns these inherent structural dependencies.

This unified multi-task prediction framework can simultaneously predict backbone atoms, Cα atoms, and amino acid types within the same network, capturing the inherent correlations between interconnected structural and sequence properties.

## Building protein structural models from deep learning predictions

The mask channels from backbone atom prediction, Cα prediction, and amino acid prediction are removed, and softmax is applied on the remaining channels to obtain backbone atom probability, Cα atom probability, and amino acid probability. These output grids for the same normalized input cryo-EM density map are then stitched back together to match the dimensions of the input map. The predictions are then utilized for building the structure of the protein in the following steps, as shown in Fig. 10C–E.

**Clustering of predicted Cα atoms.** The deep learning model usually predicts multiple spatially close voxels as Cα atoms for the same true Cα atom. To combine them, a DBSCAN[33] clustering strategy is utilized to group predicted Cα voxels whose Cα probability exceeds a threshold of 0.3 into clusters based on the proximity of their 3D coordinates (Fig. 10C). It evaluates each cluster based on the associated backbone probability scores, keeping only those clusters with average scores above 50% of the best cluster. A non-maximum suppression is applied to eliminate redundant candidates, followed by coordinate refinement based on the weighted averaging of neighboring positions. For each refined Cα candidate, its amino acid type probabilities are obtained from the deep learning prediction of amino acid types, and its neighborhood relationships with other Cα candidates according to the distance between them are established.

**Constructing the Cα backbone model from predicted Cα atoms and amino acid types.** Following the Cα clustering, EModelX(+AF)'s backbone tracing protocol[24] is used to build a Cα backbone model as shown in Fig. 10D. From the refined Cα candidates, a Cα backbone connectivity graph is constructed by connecting candidates within a 2–6 Å distance range through edges. The graph is then pruned to ensure that each node has a maximum degree of 2, thereby preserving the linear connectivity of the backbone chain. Linear backbone traces are subsequently extracted from this connectivity graph.

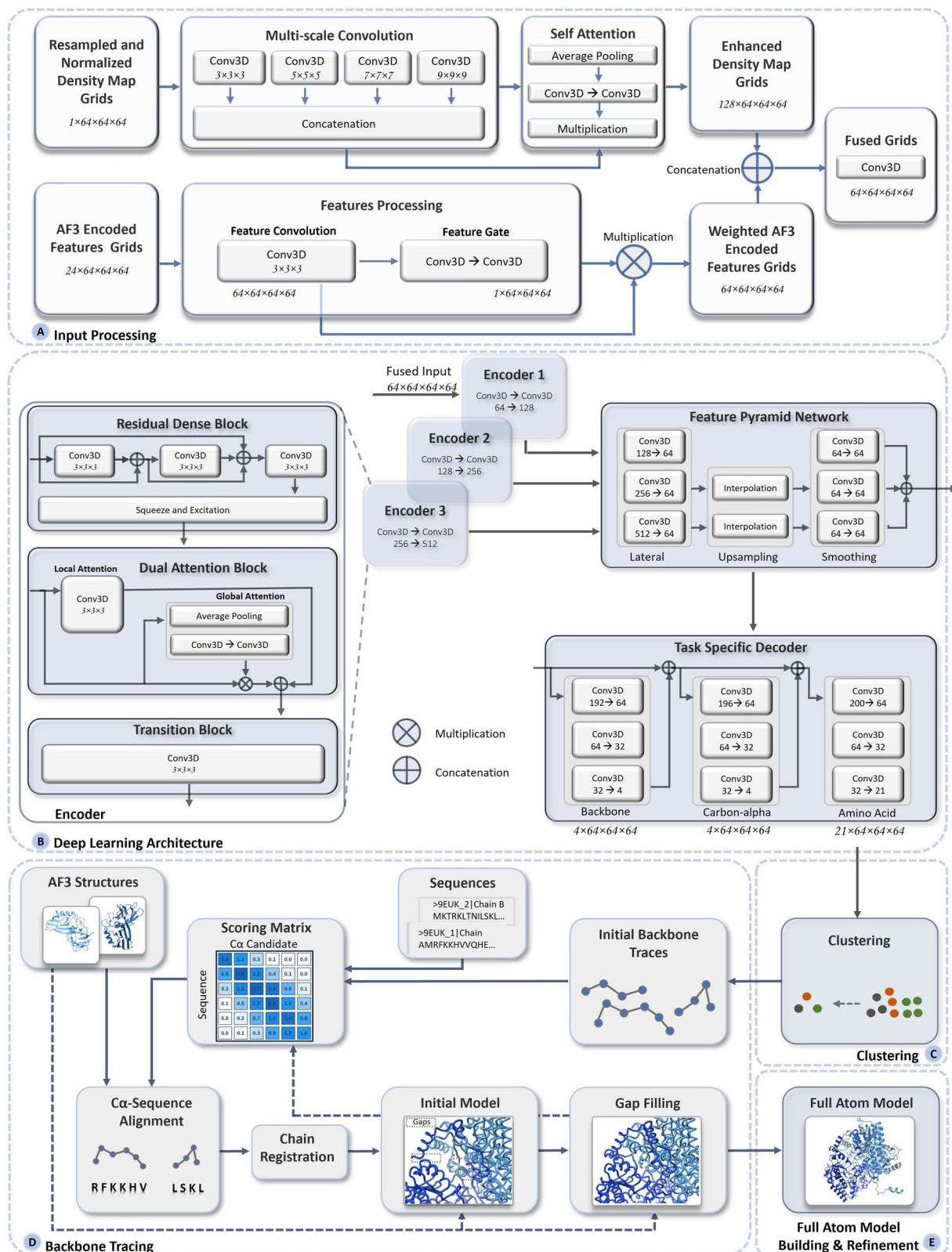

**Fig. 10 | MICA architecture for protein structure modeling from cryo-EM density maps and AF3-predicted structures.** The pipeline consists of five main stages: **A** input processing takes cryo-EM density maps and AF3-encoded features, processes them through multi-scale convolutions and attention mechanisms, and then fuses the enhanced representations. **B** Deep learning architecture uses a multi-encoder system with residual dense blocks and dual attention mechanisms to extract hierarchical features, which are then processed through an FPN and task-specific decoders to predict backbone atoms, Cα atoms, and amino acid types. **C** Clustering for identifying high-confidence Cα atoms. **D** Backbone tracing performs Ca-sequence alignment by identifying high-confidence anchor points and iteratively extending and refining the backbone trace using AF3 structures, ultimately producing complete protein backbone models through gap filling. **E** The backbone model obtained through gap filling is converted to a full-atom model and further refined.

To establish a mapping between the deep learning predictions and the sequences of protein chains, a 3D scoring matrix is constructed, containing probabilities that each Cα candidate $k$ matches the amino acid type of residue $j$ in chain sequence $i$ across all sequences. This initial scoring matrix is further enhanced by utilizing information derived from multi-step connectivity propagation between Cα candidates through N-hop steps, leading to an enhanced scoring matrix between Cα candidates and residues that enables high-confidence Cα-sequence alignment (matching).

High-scoring positions within the enhanced score matrix are identified to select initial Cα-sequence alignment fragments. These fragments then undergo bidirectional extension along both sequence and structure (Cα) dimensions to maximize coverage. When AF3-predicted structures are available for the fragments, they are incorporated to improve alignment accuracy in the regions where the deep learning predictions are poor. Overlapping fragments are then merged. The resulting fragments are ranked according to their alignment scores, with priority given to high-quality fragments that facilitate chain registration. To assign fragments to optimal matching chains, a greedy assignment algorithm is employed that prevents sequence position conflicts while maximizing overall chain coverage. In cases where AF3-predicted structures are accessible, the Cα-residue alignment undergoes iterative refinement with sequence position adjustments to minimize RMSD deviation from the AF3 structures. A Cα backbone model with some gaps is then constructed.

A gap-filling strategy is employed to use a bidirectional search that simultaneously extends fragments from both gap endpoints. AF3-predicted structures are utilized to eliminate implausible extension (threading) configurations. The gap-filling process is carried out iteratively until all identified gaps are successfully filled for each chain. The final output is a complete Cα backbone model consisting of Cα atoms and their assigned amino acid types.

**Full-atom model building and refinement.** The Cα backbone model obtained from the previous step is used by PULCHRA[27] to build a full-atom structural model, which is further refined using phenix.real_space_refine tool[28] against the input cryo-EM density map to generate the final structural model.

**Training of the deep learning model**
The multimodal deep learning model of MICA was trained to predict backbone atoms, Cα atoms, and amino acid types simultaneously using weighted cross-entropy loss with lambda regularization parameters. The overall loss function is formulated as:

$$L_{total} = \lambda_b * L_{backbone} + \lambda_c * L_{carbon\ alpha} + \lambda_a * L_{amino\ acid} \qquad (2)$$

where, $\lambda_b + \lambda_c + \lambda_a = 1$.
The weighted cross-entropy loss for each task is defined as:

$$L(x, y) = -\frac{1}{N} \sum_{n=1}^{N} \sum_{k=1}^{K} w_k * y_{n,k} * \log\left(\frac{\exp(x_{n,k})}{\sum_{i=1}^{K} \exp(x_{n,i})}\right) \qquad (3)$$

where, $L(x, y)$ is the weighted cross-entropy loss, $N$ is the number of samples in the minibatch, $K$ is the number of classes, $w_k$ denotes the weight assigned to class $k$, $x_{n,k}$ is the logit for class $k$ in sample $n$, and $y_{n,k}$ is a binary indicator that signifies whether class $k$ is the correct classification for sample $n$.

The lambda values for backbone atom prediction, Cα atom prediction, and amino acid prediction were initially set to 0.6, 0.3, and 0.1, respectively, to prioritize backbone atom prediction in early epochs, then smoothly transitioned to new weights of 0.25, 0.4, and 0.35 at epoch 25. Gradient clipping was applied to prevent gradient explosion during training, while an adaptive learning rate starting from 0.0001 was employed for optimization. On-the-fly data augmentation techniques and a light dropout of 0.01 were implemented during training to mitigate

overfitting issues. Data augmentation strategies included adding Gaussian noise, blurring, and intensity augmentation in the normalized cryo-EM density map grids, and spatial augmentation with random 90-degree rotation, flipping, and translation of normalized cryo-EM density map grids and AF3-predicted grids simultaneously. The optimal model checkpoint with the lowest validation loss from epoch 33 was saved for inference.

**Performance evaluation metrics**
The following six metrics were used to benchmark MICA and other deep learning modeling methods.

**TM-score.** The TM-score[34] evaluates the structural similarity between the predicted and true structures, normalized by the length of the true structure. This metric provides a normalized measure of how closely the predicted structure aligns with the true structure. TM-score is one of the most widely used metrics for calculating the similarity between predicted structures and true structures. It ranges from 0 (no similarity) to 1 (identical). It considers both structure match and sequence match and, therefore, is a highly stringent metric for evaluating structural models built from cryo-EM data. We used the US-align tool[35] to calculate TM-score.

**Cα match.** The Cα match is the percentage of predicted Cα atoms that align with true Cα atoms within a specified distance threshold. A higher Cα match indicates more central backbone atoms are correctly identified and positioned. This metric was computed using Phenix's chain_comparison tool[7]. This metric does not consider if Cα atoms are correctly matched with amino acids or linked into peptide chains.

**Cα quality score.** The Cα quality score[12] measures both the precision and coverage of predicted Cα atoms, compared to the true structure. Here, the precision refers to the Cα match score, and the coverage is the ratio of the number of predicted residues in a model to the number of residues in the true structure.

**Aligned Cα length.** The aligned Cα length refers to the number of Cα atoms in the predicted structure that are correctly aligned with those in the true structure. It provides insight into how completely the predicted structure matches the true structure. A higher length indicates that more residues in the protein are correctly modeled in the predicted structure. We used the US-align tool[35] to compute this metric.

**Sequence identity.** The sequence identity is conceptually similar to the sequence match. It is the percentage of identical residues between two aligned structures (a structural model and a ground truth structure), calculated as the ratio of matching positions to the alignment length. We used the US-align tool[35] to calculate this value.

**Sequence match.** The sequence match is the percentage of identical residues among aligned Cα atoms between structural models and true structures. We utilized Phenix's chain_comparison tool[7] to compute this metric.

**Reporting summary**
Further information on research design is available in the Nature Portfolio Reporting Summary linked to this article.

**Data availability**
The training dataset and test dataset are available on zenodo[36] and can be accessed at https://zenodo.org/records/15756654. The 3D cryo-EM density maps and ground truth structures for each protein presented in the manuscript were downloaded from EMDB[31] and RCSB PDB[29] websites, respectively.

## Code availability
The scripts, programs, and instructions for downloading files, processing data, and running the MICA are found in https://github.com/jianlin-cheng/MICA.

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

## Acknowledgements

This work was supported by the National Institutes of Health (NIH) (grant #: R01GM146340).

## Author contributions

J.C. conceived and supervised this research. R.G. and J.C. designed the method. R.G. implemented, trained, and tested the method. R.G. collected the data. R.G., A.D., and J.C. analyzed the data and results. R.G. drafted the manuscript. J.C. revised the manuscript.

## Competing interests
The authors declare no competing interests.
