## [Transparent Peer Review file · Communications Chemistry]

Multimodal deep learning integration of cryo-EM and AlphaFold3 for high-accuracy protein structure determination

Corresponding Author: Professor Jianlin Cheng

Version 0:

Reviewer comments:

Reviewer #1

(Remarks to the Author)

This paper presents an innovative deep learning framework, MICA, for automatically constructing high-accuracy protein structures from cryo-EM density maps. The method effectively integrates information from cryo-EM maps and AlphaFold3 (AF3)-predicted structures. Notably, it introduces AF3 predictions directly into the modeling pipeline by extracting structural features from AF3 outputs and fusing them with features derived from the density maps to predict C α atom positions and other structural information. MICA employs a multi-task encoder-decoder architecture combined with a feature pyramid network to progressively build an initial backbone model, which is further refined into atomic-level structures. The results demonstrate a significant improvement over existing methods in terms of modeling accuracy and completeness. Moreover, MICA shows strong robustness across proteins of varying sizes and map resolutions, achieving an average TM-score of 0.93, indicating its strong potential for real-world applications. However, several issues need to be addressed and clarified before publication.

(1) The authors should analyze the impact of the quality of AlphaFold3 (AF3) models on the performance of MICA. Specifically, it would be valuable to investigate how the TM-score of the input AF3 models correlates with the TM-score of the final predicted structures. Since the proposed method relies heavily on AF3 predictions, such an analysis is important to assess the robustness and generalizability of MICA under varying input quality conditions.

(2) In Figure 1 and the Input Preprocessing section, you mention the use of Merizo to divide proteins into structural domains, followed by docking each domain into the corresponding density map using phenix.dock_in_map. How much impact does the accuracy of this domain docking step have on the subsequent deep learning training process and final prediction results? If the initial AlphaFold3 (AF3) structure is inaccurate, potentially leading to incorrect docking positions, how do you handle such cases?

(3) In Figures 3 and 5, ModelAngelo shows relatively strong performance in sequence match, indicating a higher accuracy in amino acid type prediction. However, its final predicted structural models are notably less accurate. Could you provide an analysis or possible explanation for this discrepancy?

(4) In Figure 8, your method demonstrates stable performance on cryo-EM maps with resolutions ranging from 1 to 4 Å. However, can your approach still produce high-accuracy models when the map resolution exceeds 4 Å? It would be helpful to include two representative examples to illustrate the method's effectiveness in such lower-resolution scenarios.

(5) In the first subplot of Figure 8, the legend obstructs the data distribution in the lower right corner. It is recommended to move the legend to the lower left corner to improve clarity and visibility.

Reviewer #2

(Remarks to the Author)

The authors present a novel and promising approach for atomic model building from cryo-EM maps, which integrates predicted atomic models—such as those from AlphaFold3—directly into the modeling pipeline. This multimodal fusion strategy is conceptually elegant and technically well executed. The method is implemented within a deep learning framework, thoroughly tested against multiple benchmark datasets, and demonstrates clear advantages over current state-of-the-art tools. The manuscript is also well written and generally easy to follow.

The manuscript could be further improved by addressing the following minor points:

- Number of input structures: It is unclear how many predicted atomic models are typically used as input. Clarifying this would help the reader better understand the setup.

- Execution time: Given that the method appears fully automated and scalable, it would be helpful to report average or typical

execution times for various protein sizes.

- Resolution limits: While the method is designed for high-resolution maps, it would be useful to clarify its lower resolution limit and whether local resolution estimates could be incorporated to modulate prediction confidence or guide model refinement.

- Uncertainty modeling: Currently, the method outputs a single atomic model. Could the authors comment on whether the framework could be extended to generate alternative models or uncertainty estimates, especially in ambiguous or low-resolution regions?

Overall, this is an excellent contribution to the field and a strong candidate for publication after addressing these minor points.

Version 1:

Reviewer comments:

Reviewer #1

(Remarks to the Author)

The authors have satisfactorily addressed the concerns regarding the impact of AlphaFold3 (AF3) model quality on MICA's performance. The additional quantitative analysis demonstrates that MICA consistently improves structural accuracy across AF3 inputs of varying quality, highlighting its robustness and effective feature integration. The domain-wise docking strategy further provides a practical approach to leverage reliable AF3 domains while avoiding inaccurate ones. The discussion is clear, well-supported, and all previous questions have been resolved. I have no remaining concerns and recommend acceptance.

Reviewer #2

(Remarks to the Author)

The authors have addressed all my comments. From my point of view, the paper is ready to be published.

Response to Reviewers' comments

Reviewer #1

This paper presents an innovative deep learning framework, MICA, for automatically constructing high-accuracy protein structures from cryo-EM density maps. The method effectively integrates information from cryo-EM maps and AlphaFold3 (AF3)-predicted structures. Notably, it introduces AF3 predictions directly into the modeling pipeline by extracting structural features from AF3 outputs and fusing them with features derived from the density maps to predict $C\alpha$ atom positions and other structural information. MICA employs a multi-task encoder-decoder architecture combined with a feature pyramid network to progressively build an initial backbone model, which is further refined into atomic-level structures. The results demonstrate a significant improvement over existing methods in terms of modeling accuracy and completeness. Moreover, MICA shows strong robustness across proteins of varying sizes and map resolutions, achieving an average TM-score of 0.93, indicating its strong potential for real-world applications. However, several issues need to be addressed and clarified before publication.

Response:

We thank you for recognizing MICA as an innovative framework with strong real-world potential and for appreciating our multimodal approach that integrates AlphaFold3 predictions with cryo-EM density maps.

We greatly appreciate your constructive feedback, which helps strengthen our manuscript significantly. We have addressed each of your comments as follows:

- (1) The authors should analyze the impact of the quality of AlphaFold3 (AF3) models on the performance of MICA. Specifically, it would be valuable to investigate how the TM-score of the input AF3 models correlates with the TM-score of the final predicted structures. Since the proposed method relies heavily on AF3 predictions, such an analysis is important to assess the robustness and generalizability of MICA under varying input quality conditions.

Response:

Thank you for the great suggestion about how the quality of AlphaFold3 (AF3) input models affects MICA's performance. According to your suggestion, we conducted a comprehensive analysis to address this question by comparing TM-scores between the input AF3 models and the final structural models built by MICA.

Specifically, we divided the 160 density maps from both the Cryo2StructData test dataset and Standard test dataset into three different categories as follows:

- Low Quality AF3: 39 density maps with AF3 models having TM-score less than 0.5
- Medium Quality AF3: 38 density maps with AF3 models having TM-score greater or equal to 0.5 and less than 0.8
- High Quality AF3: 83 density maps with AF3 models having TM-score greater or equal to 0.8

We then calculated the average TM-score for AF3 models in each group above and compared with the average TM-score of the final structural models built by MICA. The average TM-score comparison results are presented in the Figure 1 below:

Figure 1: Average TM-score comparison between MICA models and AF3 predicted input models of varying quality.

The results demonstrate that while MICA's performance correlates with AlphaFold3 quality, it maintains robust modeling capabilities across all AF3 model quality categories. For low-quality AF3 models (TM-score 0.32), MICA achieves significantly higher performance (0.77), indicating its ability to leverage cryo-EM data effectively even when AF3 structural predictions are unreliable. With medium-quality AF3 models (TM-score 0.62), MICA performance increases substantially (TM-score 0.89), while with high-quality AF3 models (TM-score 0.95) as input, it achieves near-optimal MICA performance (TM-score 0.96).

This continued improvement reflects MICA's intelligent feature integration architecture. The feature gate module automatically suppresses unreliable AF3 predictions while amplifying high-confidence structural information. Additionally, the random feature masking strategy is employed during training, where for each iteration, a random number is generated and compared against a threshold of 0.4. If the random number is below the threshold, the model uses only cryo-EM features, otherwise it uses combined features. This ensures MICA avoids excessive reliance on AF3 predictions and maintains robust performance regardless of structural prediction availability or quality. This performance improvement across AF3 quality categories demonstrates that MICA effectively prioritizes cryo-EM data as its core information source, utilizing AF3 features as supplementary structural guidance rather than critical dependencies.

We have added a new subsection “**Impact of AlphaFold3 model quality on MICA’s performance**” in the “Results” section of the manuscript to discuss these new results.

- (2) In Figure 1 and the Input Preprocessing section, you mention the use of Merizo to divide proteins into structural domains, followed by docking each domain into the corresponding density map using phenix.dock_in_map. How much impact does the accuracy of this domain docking step have on the subsequent deep learning training process and final prediction results? If the initial AlphaFold3 (AF3) structure is inaccurate, potentially leading to incorrect docking positions, how do you handle such cases?

Response:

Thank you for this important question. Our approach handles docking failures gracefully through a domain-wise strategy. When Phenix.dock_in_map successfully docks AlphaFold3 (AF3) domains (divided via

Merizo) into density maps, AF3 features are integrated with cryo-EM features. If docking fails for any domain, that domain's AF3 features are excluded, and the model relies solely on cryo-EM features for the corresponding density region. This allows partial utilization of reliable structural information even when the overall AF3 structure contains inaccuracies.

Importantly, even if the initial overall AF3 structure is inaccurate, dividing it into domains using Merizo may often yield some good domain structures that fit well into the density map and contribute valuable structural features. The poorly predicted domain structures are naturally filtered out during the docking process, as they typically fail to dock properly into the density map due to structural incompatibility. When docking fails for these problematic domains, the model automatically excludes their AF3 features and relies solely on cryo-EM features to make predictions for those regions. This domain-wise approach thus provides a built-in quality control mechanism that selectively incorporates only the reliable portions of AF3 predictions.

During training, incorrect AF3 predictions or improper docking serve as implicit data augmentation, forcing MICA to learn robust feature representations that don't simply memorize AF3 structural information but instead learn to effectively combine and validate features from multiple sources.

To assess the impact of docking quality on model performance, during the revision, we analyzed the 160 density maps from both the Cryo2StructData test dataset and Standard test dataset, dividing them into two groups:

- Fully Docked: 150 density maps where all AF3 domains were successfully docked
- Partially Docked: 30 density maps where not all AF3 domains could be docked

We calculated the average TM-scores for both the initial AF3 models and MICA's predictions for each group as shown in Figure 2 below. For the fully docked cases, AF3 models achieved an average TM-score of 0.74 compared to MICA's 0.91. For partially docked cases, AF3 models achieved an average TM-score of 0.64 against MICA's 0.86. The reduced performance in partially docked cases demonstrates some correlation between docking success and structural prediction quality. However, MICA's decreased performance in partially docked cases may also be attributed to lower cryo-EM density resolution in certain regions, which inherently complicates both docking and structure prediction beyond just the absence of AF3 features. Despite these challenges, MICA maintains robust performance for partially docked cases by leveraging its cryo-EM-based foundation to compensate for unreliable or missing AF3 information.

Figure 2: Performance comparison of MICA to assess the impact of docking quality.

We have added a new subsection “**Impact of domain docking accuracy on MICA’s performance**” in the “Results” section of the manuscript to discuss the new results.

- (3) In Figures 3 and 5, ModelAngelo shows relatively strong performance in sequence match, indicating a higher accuracy in amino acid type prediction. However, its final predicted structural models are notably less accurate. Could you provide an analysis or possible explanation for this discrepancy?

Response:

Thank you for your insightful comment. This phenomenon occurs because sequence match score is the percentage of identical residues among aligned C α atoms (# identical residues / aligned Ca length). ModelAngelo adopts a conservative approach, generating less complete structural models with fewer C α atoms but with higher local accuracy, which inflates its sequence match scores since fewer aligned residues create a smaller denominator in the percentage calculation. In contrast, MICA pursues a more comprehensive modeling strategy, attempting to reconstruct larger protein portions and generate substantially more complete models with more C α atoms. This is the reason MICA’s models have a substantially greater aligned C α length (average: 1396.81) than ModelAngelo’s (average: 1191.74), but MICA’s sequence match score (96.07%) is only slightly lower than ModelAngelo’s (98.33%) on the Cryo2StructData test dataset. Considering both aligned C α length (coverage) and sequence match/identity (precision), MICA outperforms ModelAngelo substantially. For instance, MICA achieves a much higher TM-score (0.92) than ModelAngelo’s (0.75) on the test dataset because this metric considering both the completeness of the models (coverage) and the sequence match (precision).

Here are some examples illustrating this phenomenon. For protein complex EMD-28866 with 5,533 total residues, ModelAngelo predicted 1,666 residues and aligned 899 C α atoms, achieving a sequence match score of 90% (approximately 809 identical residues), while MICA predicted 4,577 residues, aligned 3,630 C α atoms, and achieved a sequence match score of 63.6% (approximately 2,287 identical residues). Similarly, for EMD-15685 with 1,156 total residues, ModelAngelo predicted 828 residues, aligned 535 C α atoms with a 98.8% sequence match score (approximately 528 identical residues), whereas MICA predicted 1,070 residues, aligned 963 C α atoms with a 91.9% sequence match score (approximately 884 identical residues). Despite ModelAngelo's higher sequence match score in both cases, MICA's models contain significantly more identical residues with the true structures, demonstrating how MICA's ambitious reconstruction approach yields substantially more complete structural models.

The following text has been added into the main manuscript in “MICA outperforms ModelAngelo and EModelX(+AF) on the Cryo2StructData test dataset” subsection of the “Results” to explain it:

"The higher sequence match of ModelAngelo is due to its structural models being generally much less complete than those of MICA and EModelX(+AF), as indicated by its much shorter aligned Ca length. Since sequence match is the percentage of identical residues within aligned Ca atoms (# identical residues / aligned Ca length), ModelAngelo's fewer aligned atoms create a smaller denominator, leading to higher sequence match scores despite of less complete structural coverage."

- (4) In Figure 8, your method demonstrates stable performance on cryo-EM maps with resolutions ranging from 1 to 4 Å. However, can your approach still produce high-accuracy models when the map resolution

exceeds 4 Å? It would be helpful to include two representative examples to illustrate the method's effectiveness in such lower-resolution scenarios.

Response:

Thank you for this important question regarding MICA's performance on lower-resolution cryo-EM maps.

To answer this question, we evaluated MICA on 8 representative density maps with resolutions ranging from 4-6 Å, all recently released after January 1, 2025, on the EMDB website. The results, presented in Table 1 below, reveal a mixed performance pattern that reflects both the opportunity and challenges associated with lower-resolution structural modeling. For density maps such as EMD-43679, EMD-51169, and EMD-45739, the MICA's modeling accuracy was suboptimal, likely due to insufficient density details for reliable atomic-level predictions, but for several other cases like EMD-62354 and EMD-49996, MICA produced reasonably accurate models.

This varied performance at resolutions exceeding 4 Å is expected, as the reduced density detail at lower resolutions makes it increasingly difficult to distinguish individual atoms and side chain orientations, which are critical for accurate atomic model building. These results highlight that while MICA maintains some modeling capability beyond its optimal resolution range (< 4 Å), users should be careful when applying it to maps with resolutions significantly exceeding 4 Å.

Moreover, we compared MICA with ModelAngelo and EModelX(+AF) on the low-resolution maps. MICA demonstrated superior performance over ModelAngelo and EModelX(+AF), which either failed to generate complete models or produced only fragmentary structures for these lower-resolution density maps (see Supplementary Table S4), highlighting MICA's comparative advantage even when operating beyond its optimal resolution range.

Table 1: Performance evaluation of MICA on low-resolution cryo-EM density maps with resolutions ranging from 4-6 Å.

EMD ID	PDB ID	Resolution (Å)	TM-Score	C α Match	C α Quality Score	Model Length	Reference Length	Aligned Length	Sequence Identity	Sequence Match
62354	9kht	4.85	0.789	72.9	60.195	2260	2737	2222	0.939	87.3
49996	9o13	5.8	0.897	85.4	136.689	1671	1044	983	0.893	75.3
44981	9bvw	5.1	0.717	69.3	60.348	1773	2036	1560	0.851	74.7
45739	9cm5	4.61	0.58	52.9	43.621	1589	1927	1390	0.668	29.6
43679	8vyv	5.86	0.268	35.2	15.421	276	630	193	0.596	56.3
46873	9dhq	4.78	0.629	77.8	56.880	1702	2328	1523	0.909	84.1
51169	9ga2	4.9	0.484	75.7	86.804	1540	1343	730	0.804	60.3
61490	9jhs	5.02	0.691	75.8	68.163	1446	1608	1255	0.72	69.3

We have added a new subsection “**MICA's performance on cryo-EM density maps with lower (4-6 Å) resolution**” in the “Results” section of the manuscript to discuss the new results.

- (5) In the first subplot of Figure 8, the legend obstructs the data distribution in the lower right corner. It is recommended to move the legend to the lower left corner to improve clarity and visibility.

Response:

Thank you for the great suggestion. We have moved the legend to the lower left corner as recommended, which provides much better visibility of the complete data distribution while maintaining figure clarity.

Reviewer #2:

The authors present a novel and promising approach for atomic model building from cryo-EM maps, which integrates predicted atomic models—such as those from AlphaFold3—directly into the modeling pipeline. This multimodal fusion strategy is conceptually elegant and technically well executed. The method is implemented within a deep learning framework, thoroughly tested against multiple benchmark datasets, and demonstrates clear advantages over current state-of-the-art tools. The manuscript is also well written and generally easy to follow.

The manuscript could be further improved by addressing the following minor points:

Response:

Thank you very much for the encouraging assessment of our multimodal fusion strategy and comprehensive evaluation. The positive feedback on both the technical contribution and manuscript quality is much appreciated. We are grateful for the constructive suggestions provided to further improve the manuscript and have carefully addressed each of the four points raised.

Below, we provide detailed responses to each of the specific points:

- Number of input structures: It is unclear how many predicted atomic models are typically used as input. Clarifying this would help the reader better understand the setup.

Response:

Thank you for your great question. For each protein target, AlphaFold3 (AF3) generates structures on a per-chain basis. Specifically, the AlphaFold3 server is used to predict 5 different models (ranked 0-4) for each unique chain in the protein from its FASTA sequence, from which we select model 0 with the highest confidence score for the chain as input. Therefore, the total number of structural models from AlphaFold3 predictions equals the number of unique chains in the protein.

These AF3 chain models are subsequently divided into structural domains using Merizo. Domain counts can vary significantly between different chains depending on their structural complexity. This domain-wise approach means that for a protein complex with N unique chains, MICA processes N AF3 models that are decomposed into a variable number of domains.

These domains are finally docked to cryo-EM density map using Phenix.dock_in_map tool and AlphaFold3 encodings are generated, which are utilized by MICA.

The following text has been added in the “Input preprocessing and label preparation” subsection of “Materials and Methods” section of the manuscript to explain this.

“For each cryo-EM density map, AlphaFold3 was used to predict structures for each chain from the protein's FASTA sequence. The AlphaFold3 server generates 5 different models (ranked 0-4) for each unique chain, from which we selected model 0 with the highest confidence score. Thus, the total number of atomic models from AlphaFold3 predictions corresponds to the number of unique chains in the protein complex. These selected atomic models were then divided into structural domains using Merizo and subsequently docked into the density map using the phenix.dock_in_map command to align them within the same 3D coordinate system as the cryo-EM data.”

- Execution time: Given that the method appears fully automated and scalable, it would be helpful to report average or typical execution times for various protein sizes.

Response:

Thank you for this valuable suggestion. We have conducted a comprehensive time analysis across 18 protein structures of varying sizes and complexities. The execution time is presented in Table 2 below.

Table 2: Typical Execution time for different sized proteins

EMD ID	PDB ID	Original Map Shape	Normalized Map Shape	Number of Residues	Number of Domains	AF3 Results Processing Time (min)	Domains Docking Time (min)	Atomic Model Building Time (min)	Total Time (min)
26993	8ctk	288x288x288	209x209x209	376	4	0.16	14.83	1.81	16.8
14716	7zh0	350x350x350	231x231x231	495	3	0.1	9.13	1.4	10.63
28660	8exr	300x300x300	231x231x231	590	11	0.3	189.48	5.85	195.63
15673	8aur	280x280x280	306x306x306	699	3	0.26	26.58	2.4	29.24
27656	8dql	300x300x300	330x330x330	771	9	0.26	82.55	3.56	86.37
14066	7qla	180x180x180	196x196x196	874	6	0.2	30.58	2.58	33.36
26978	8ct2	344x344x344	143x143x143	984	7	0.33	56.68	1.08	58.09
33233	7xjp	240x240x240	260x260x260	1089	6	0.2	25.36	3.01	28.57
26974	8csx	264x264x264	110x110x110	1158	6	0.26	80.4	1.95	82.61
27138	8d1v	300x300x300	249x249x249	1281	5	0.15	64.78	3.31	68.24
34023	7yqc	240x240x240	250x250x250	1386	7	0.6	26.6	1.93	29.13
15635	8at6	320x320x320	352x352x352	1580	8	0.53	32.81	5.26	38.6
14842	7zny	224x224x224	289x289x289	1879	11	0.4	153.9	6.78	161.08
27761	8dwv	300x300x300	423x423x423	2084	12	0.55	175.46	9.8	185.81
15540	8ane	350x350x350	353x353x353	2278	17	0.61	246.23	6.11	252.95
27760	8dwu	270x270x270	350x350x350	2506	19	0.81	234.35	6.93	242.09
15378	8ae1	320x320x320	349x349x349	2859	12	0.36	294.01	9.16	303.53
34738	8hgg	256x256x256	269x269x269	3238	12	0.48	201.93	9.05	211.46

Our benchmarking reveals that total processing time ranges from 10.63 minutes for small proteins (495 residues, 3 domains) to 303.53 minutes for very large proteins (2859 residues, 12 domains). The relationship between protein size and execution time is non-linear, primarily driven by domain complexity rather than residue count alone. Domain docking consistently represents the computational bottleneck, accounting for 85-97% of total execution time across all test cases. For example, in EMD-34738 (3238 residues, 12 domains), domain docking required 201.93 minutes out of 211.46 total minutes. This bottleneck occurs because the domain docking step is performed through Phenix using CPU-based computation rather than GPU acceleration, making it the most time-intensive component of the pipeline. Based on the data in Table 2, we estimate the typical execution times for different sized proteins as follows:

- **Proteins with 300-700 residues:** 10-30 minutes average
- **Proteins with 700-1400 residues:** 30-90 minutes average
- **Proteins with 1400-2500 residues:** 150-240 minutes average
- **Proteins with 2500 + residues:** 240-300+ minutes average

All benchmarks were carried out using 24 CPU cores on one NVIDIA A100 GPU with 80GB of memory.

We have added a “**Typical execution time**” subsection in “**Results**” section of the main manuscript as follows to discuss the new results.

- Resolution limits: While the method is designed for high-resolution maps, it would be useful to clarify its lower resolution limit and whether local resolution estimates could be incorporated to modulate prediction confidence or guide model refinement.

Response:

Thank you for this important question regarding MICA's performance on lower-resolution cryo-EM maps. MICA is primarily optimized for high-resolution density maps in the 1-4 Å range, where it demonstrates consistently stable performance. While MICA can process density maps with resolutions exceeding 4 Å, the model accuracy may be variable due to the inherent limitations of lower-resolution data. To address your question, we evaluated MICA on 8 representative density maps with resolutions ranging from 4-6 Å, all recently released after January 1, 2025, on the EMDB website to determine the practical lower resolution limit and characterize performance degradation patterns. The results, presented in Table 3 below, reveal a mixed performance pattern that reflects the opportunity and challenges associated with lower-resolution structural modeling. For density maps such as EMD-43679, EMD-51169, and EMD-45739, the modeling accuracy was suboptimal, likely due to insufficient density detail for reliable atomic-level predictions. but, for several other cases like EMD-62354 and EMD-49996, MICA produced reasonably accurate models.

This varied performance at resolutions exceeding 4 Å is expected, as the reduced density detail at lower resolutions makes it increasingly difficult to distinguish individual atoms and side chain orientations, which are critical for accurate atomic model building. These results highlight that while MICA maintains some capability beyond its optimal resolution range, users should be careful when applying it to maps with resolutions significantly exceeding 4 Å.

Moreover, MICA demonstrated superior robustness compared to ModelAngelo and EModelX(+AF) on these low-resolution density maps, which either failed to generate complete models or produced only fragmentary structures for these lower-resolution density maps (see Supplementary Table S4), highlighting MICA's comparative advantage even when operating beyond its optimal resolution range.

Table 3: Performance evaluation of MICA on low-resolution cryo-EM density maps with resolutions ranging from 4-6 Å.

EMD ID	PDB ID	Resolution (Å)	TM-Score	C α Match	C α Quality Score	Model Length	Reference Length	Aligned Length	Sequence Identity	Sequence Match
62354	9kht	4.85	0.789	72.9	60.195	2260	2737	2222	0.939	87.3
49996	9o13	5.8	0.897	85.4	136.689	1671	1044	983	0.893	75.3
44981	9bww	5.1	0.717	69.3	60.348	1773	2036	1560	0.851	74.7
45739	9cm5	4.61	0.58	52.9	43.621	1589	1927	1390	0.668	29.6
43679	8vyv	5.86	0.268	35.2	15.421	276	630	193	0.596	56.3
46873	9dhq	4.78	0.629	77.8	56.880	1702	2328	1523	0.909	84.1
51169	9ga2	4.9	0.484	75.7	86.804	1540	1343	730	0.804	60.3
61490	9jhs	5.02	0.691	75.8	68.163	1446	1608	1255	0.72	69.3

Regarding incorporation of local resolution information, we are planning its integration as a key enhancement for future MICA versions through a multi-faceted approach. This strategy combines resolution-aware feature weighting, where local resolution estimates serve as an additional input channel to dynamically modulate cryo-EM feature importance with high-resolution regions receiving full weights and lower-resolution areas getting reduced emphasis, coupled with confidence-calibrated scoring that generates per-residue uncertainty estimates based on local resolution quality metrics. Additionally, we will implement resolution-guided AF3 integration where lower-resolution regions receive progressively increased weighting from AF3 structural priors to ensure physically reasonable predictions when experimental constraints are weak. This integrated framework enables MICA to maintain high precision in well-resolved regions while providing robust, confidence-quantified predictions in challenging areas, ultimately delivering more reliable and interpretable structural models that appropriately balance experimental evidence with prior knowledge across heterogeneous cryo-EM datasets.

In the revised manuscript, we have added a new subsection “**MICA’s performance on cryo-EM density maps with lower (4-6 Å) resolution**” in the “Results” section of the manuscript to discuss the new results.

Regarding your suggestion on local resolution estimation incorporation, we have added following text in the “Discussion” section of the revised manuscript.

“Another important enhancement involves developing local resolution integration capabilities through a multi-faceted approach. This framework will combine resolution-aware feature weighting, where local resolution estimates provided as an additional input dynamically modulate cryo-EM feature importance, with confidence-calibrated scoring that generates per-residue uncertainty estimates based on local

resolution quality metrics and geometric consistency between experimental and computational features. The system will also integrate resolution-guided AF3 integration that progressively increases structural prior weighting in lower-resolution regions to ensure physically reasonable predictions when experimental constraints are weak. This integrated approach will enable MICA to maintain high precision in well-resolved regions while providing robust, confidence-quantified alternative models in challenging areas, ultimately delivering more reliable and interpretable structural models across heterogeneous cryo-EM datasets.”

- Uncertainty modeling: Currently, the method outputs a single atomic model. Could the authors comment on whether the framework could be extended to generate alternative models or uncertainty estimates, especially in ambiguous or low-resolution regions?

Response:

Thank you for this insightful question. Currently, MICA outputs a single atomic model representing the most likely structural prediction. However, we recognize the significant value of extending the framework to generate uncertainty estimates and alternative models, particularly for ambiguous or low-resolution regions where structural interpretation is inherently challenging, and we plan to extend MICA in the next version.

The extended framework will use local resolution estimates as an additional input which evaluates map reliability and assign confidence scores ranging from high precision in well-resolved areas (2-3 Å) to enhanced uncertainty quantification in lower-resolution regions (>5 Å). The system will incorporate confidence-calibrated scoring that generates per-residue uncertainty estimates based on local data quality, geometric consistency, and agreement between experimental and predicted features. Additionally, the framework will integrate uncertainty-aware structural prior weighting that selectively incorporates AlphaFold3 predictions as structural guidance in regions where experimental density is weak or ambiguous, while clearly distinguishing and reporting the relative contributions of experimental evidence versus computational predictions to ensure users understand which portions of the final model are primarily data-driven versus prior-informed. This integrated approach will enable MICA to maintain high precision predictions in well-resolved regions while providing robust, confidence-quantified alternative models in challenging areas, ultimately delivering more reliable and interpretable atomic models across heterogeneous cryo-EM datasets.

Accordingly, the following text has been added in the “Discussion” section of the revised manuscript as future work:

“Another important enhancement involves developing local resolution integration capabilities through a multi-faceted approach. This framework will combine resolution-aware feature weighting, where local resolution estimates provided as an additional input dynamically modulate cryo-EM feature importance, with confidence-calibrated scoring that generates per-residue uncertainty estimates based on local resolution quality metrics and geometric consistency between experimental and computational features. The system will also integrate resolution-guided AF3 integration that progressively increases structural prior weighting in lower-resolution regions to ensure physically reasonable predictions when experimental constraints are weak. This integrated approach will enable MICA to maintain high precision in well-resolved regions while providing robust, confidence-quantified alternative models in challenging areas,

ultimately delivering more reliable and interpretable structural models across heterogeneous cryo-EM datasets.”